# MATRIX SKETCHING IN BANDITS: CURRENT PITFALLS AND NEW FRAMEWORK

## ABSTRACT

The utilization of sketching techniques has progressively emerged as a pivotal method for enhancing the efficiency of online learning. In linear bandit settings, current sketch-based approaches leverage matrix sketching to reduce the per-round time complexity from $\Omega\left(d^2\right)$ to $O(d)$, where $d$ is the input dimension. Despite this improved efficiency, these approaches encounter critical pitfalls: if the spectral tail of the covariance matrix does not decrease rapidly, it can lead to *linear regret*. In this paper, we revisit the regret analysis and algorithm design concerning approximating the covariance matrix using matrix sketching in linear bandits. We illustrate how inappropriate sketch sizes can result in unbounded spectral loss, thereby causing linear regret. To prevent this issue, we propose Dyadic Block Sketching, an innovative streaming matrix sketching approach that adaptively manages sketch size to constrain global spectral loss. This approach effectively tracks the best rank-$k$ approximation in an online manner, ensuring efficiency when the geometry of the covariance matrix is favorable. Then, we apply the proposed Dyadic Block Sketching to linear bandits and demonstrate that the resulting bandit algorithm can achieve *sublinear regret* without prior knowledge of the covariance matrix, even under the worst case. Our method is a general framework for efficient sketch-based linear bandits, applicable to all existing sketch-based approaches, and offers improved regret bounds accordingly. Additionally, we conduct comprehensive empirical studies using both synthetic and real-world data to validate the accuracy of our theoretical findings and to highlight the effectiveness of our algorithm.

## 1 INTRODUCTION

The Multi-Armed Bandits (MAB) model represents a framework for sequential decision-making under conditions of partial information (Robbins, 1952). In each round, the player selects one of the $K$ arms to maximize cumulative rewards. The player's strategy, which guides action choices based on previous observations, is referred to as a policy. The player's objective is to develop a policy that minimizes regret, which is defined as the difference between the total reward of the optimal policy and that of the chosen policy.

We consider the Stochastic Linear Bandit (SLB) model, a variant of the MAB model under a linear assumption (Abbasi-Yadkori et al., 2011; Auer, 2002; Chu et al., 2011; Dani et al., 2007). In SLB, at round $t$, the player selects a arm $\boldsymbol{x}_t$ from an alternate set $\mathcal{X}$, and then observes the reward $r_t$. The expected reward $E[r_t|\boldsymbol{x}_t] = \boldsymbol{x}_t^\top \boldsymbol{\theta}_\star$, where $\boldsymbol{\theta}_\star$ represents unknown coefficients. The player's objective is to minimize the regret over the total $T$ rounds, defined as:

$$\text{Regret}_T = \sum_{t=1}^{T} \max_{\boldsymbol{x} \in \mathcal{X}} \boldsymbol{x}^\top \boldsymbol{\theta}_\star - \sum_{t=1}^{T} \boldsymbol{x}_t^\top \boldsymbol{\theta}_\star. \tag{1}$$

Utilizing upper confidence bounds and the regularized least squares estimator, Abbasi-Yadkori et al. (2011) introduced the well-known OFUL algorithm, which achieves a regret of $\tilde{O}(d\sqrt{T})$ and exhibits a computational complexity of $\Omega(d^2)$, where $d$ represents the dimension of the data. However, in real-world decision-making problems, the data dimension $d$ often increases rapidly, making traditional bandit algorithms excessively time-consuming (Calandriello et al., 2019; Xu et al., 2020; Deshpande & Montanari, 2012; Zhang et al., 2024). To address this issue and eliminate

the quadratic dependence on $d$, various approaches have adopted streaming sketching methods to enhance efficiency. Yu et al. (2017) employ random projection to map high-dimensional arms to a low-dimensional subspace. Another line of these works is based on a well-known deterministic sketching algorithm—Frequent Directions (FD), which has been proven to offer better theoretical guarantees than random projection under the streaming setting (Liberty, 2013; Woodruff et al., 2014; Ghashami et al., 2016). Kuzborskij et al. (2019) directly use FD to sketch covariance matrices of linear bandits, successfully reducing time complexity from $\Omega(d^2)$ to $O(dl + l^2)$ while achieving an upper regret bound of $\tilde{O}\left((1 + \Delta_T)^{\frac{3}{2}}(l + d\log(1 + \Delta_T))\sqrt{T}\right)$, where $l < d$ is the sketch size and $\Delta_T$ represents the spectral error caused by the shrinking of FD. Building on this foundation, Chen et al. (2021) introduce Robust Frequent Directions (RFD) to reduce the order of $\Delta_T$ and decouple $d$ and $\Delta_T$, achieving an improved regret bound of $\tilde{O}\left((\sqrt{l + d\log(1 + \Delta_T)} + \sqrt{\Delta_T})\sqrt{lT}\right)$.

Despite recent advancements, matrix sketching in linear bandits still faces several pitfalls in practical applications (Kuzborskij et al., 2019; Chen et al., 2021; Calandriello et al., 2019). Upon careful examination of the previous regret bounds, the $\sqrt{T}$ term is associated with a spectral error $\Delta_T$, which arises from the discrepancy between the sketched and non-sketched regularized least squares estimators. The magnitude of the spectral error $\Delta_T$ relates to the fixed sketch size $l$ and the spectral tail of the covariance matrix, implying that a slow decrease in this tail can contribute to linear regret. In practice, this is evident as an inappropriate sketch size can significantly hinder the performance of online learning algorithms. Since the spectral information of the covariance matrix is unknown prior to online learning, selecting an optimal pre-set, fixed sketch size is challenging. This raises a natural question: *Can we adaptively adjust the sketch size in an online manner to avoid the pitfall of linear regret in current methods?*

In this paper, we demonstrate that the answer is "yes" by developing a novel framework for efficient sketch-based linear bandit algorithms. Specifically, this work makes three key contributions:

• We revisit the fundamental problem of approximating the covariance matrix through matrix sketching. We analyze the critical condition for linear regret in sketch-based methods, which depends on unpredictable properties of the streaming matrix. From both theoretical and experimental perspectives, we demonstrate that the inability of previous methods to avoid the pitfall of linear regret stems from the difficulty of pre-setting an appropriate fixed sketch size.

• We propose Dyadic Block Sketching, a multi-scale matrix sketching method that imposes a constraint on the global spectral error by managing the error bound within each block. We prove that the cumulative spectral error upper bound from Dyadic Block Sketching conforms to a specified error $\epsilon$. This approach allows the sketch size to be dynamically adjusted to accommodate the given error, even without prior knowledge of the matrix structure. Additionally, we demonstrate that Dyadic Block Sketching effectively tracks the best rank-$k$ approximation in the streaming setting, aligning with the performance of a single deterministic sketch.

• We introduce an efficient framework for sketch-based linear bandits using Dyadic Block Sketching, effectively addressing the pitfall of linear regret in previous works. Our framework is robust, scalable, and capable of achieving various regret bounds through different sketching techniques. By tracking the best rank-$k$ approximation, our method can significantly reduce the computational cost of linear bandits when the covariance matrix has favorable properties.

**Related Work.** Two classes of prior work are particularly relevant to our study: matrix sketching algorithms in the unbounded streaming model and sketch-based online learning algorithms. Streaming matrix sketching methods can be broadly categorized into three groups: The first approach is sampling a small subset of matrix rows or columns that approximates the entire matrix (Deshpande & Rademacher, 2010; Frieze et al., 2004). The second approach is randomly combining matrix rows via random projection. Several results are available in the literature, including random projections and hashing (Sarlos, 2006; Achlioptas, 2001). The third approach employs a deterministic matrix sketching technique proposed by Liberty (2013), which adapts the well-known MG algorithm from Misra & Gries (1982) (originally used for approximating item frequencies) to sketch a streaming matrix by tracking its frequent directions. For further details, we refer readers to the survey by Woodruff et al. (2014). In sketch-based online learning algorithms, most existing work aims to enhance efficiency through sketching. Beyond linear bandits setting, matrix sketching is also employed to accelerate second-order online gradient descent (Luo et al., 2016), online kernel

learning (Calandriello et al., 2017; Luo et al., 2019), stochastic optimization (Gonen et al., 2016), and contextual batched bandits (Zhang et al., 2024).

**Organization.** The remainder of this paper is structured as follows: Section 2 revisits matrix sketching methods in linear bandits and highlights the current pitfalls. Section 3 presents a novel multi-scale sketching method for achieving a constrained global error bound. Section 4 introduces a new framework for efficient sketch-based linear bandits. Section 5 provides a detailed report of the experimental results. Finally, Section 6 concludes the paper and offers a discussion. All proofs and additional algorithmic details are provided in the appendices.

## 2 REVISITING MATRIX SKETCHING IN LINEAR BANDITS

**Notations.** Let $[n] = \{1, 2, \ldots, n\}$, upper-case bold letters (e.g., $\boldsymbol{A}$) represent matrix and lower-case bold letters (e.g., $\boldsymbol{a}$) represent vectors. We denote by $\|\boldsymbol{A}\|_2$ and $\|\boldsymbol{A}\|_F$ the spectral and Frobenius norms of $\boldsymbol{A}$. We define $|\boldsymbol{A}|$ and $\mathrm{Tr}(\boldsymbol{A})$ as the determinant and trace of matrix $\boldsymbol{A}$. For a positive semi-definite matrix $\boldsymbol{A}$, the matrix norm of vector $\boldsymbol{x}$ is defined by $\|\boldsymbol{x}\|_{\boldsymbol{A}} = \sqrt{\boldsymbol{x}^\top \boldsymbol{A} \boldsymbol{x}}$. For two positive semi-definite matrices $\boldsymbol{A}$ and $\boldsymbol{B}$, we use $\boldsymbol{A} \succeq \boldsymbol{B}$ to represent the fact that $\boldsymbol{A} - \boldsymbol{B}$ is positive semi-definite. We use $\boldsymbol{A} = \boldsymbol{U}\boldsymbol{\Sigma}\boldsymbol{V}^\top$ to represent the SVD of $\boldsymbol{A}$, where $\boldsymbol{U}, \boldsymbol{V}$ denote the left and right matrices of singular vectors and $\boldsymbol{\Sigma} = \mathrm{diag}[\sigma_1, \ldots, \sigma_n]$ is the diagonal matrix of singular values in the descending order. We define $\boldsymbol{A}_{[k]} = \boldsymbol{U}_k \boldsymbol{\Sigma}_k \boldsymbol{V}_k^\top$ for $k \leq \mathrm{rank}(\boldsymbol{A})$ as the best rank-$k$ approximation to $\boldsymbol{A}$, where $\boldsymbol{U}_k \in \mathbb{R}^{n \times k}$ and $\boldsymbol{V}_k \in \mathbb{R}^{d \times k}$ are the first $k$ columns of $\boldsymbol{U}$ and $\boldsymbol{V}$.

### 2.1 LINEAR BANDITS THROUGH MATRIX SKETCHING

Within the linear bandit setting, the reward for choosing action $\boldsymbol{x}_t$ is defined as $r_t = \boldsymbol{x}_t^\top \boldsymbol{\theta}_\star + z_t$, where $\boldsymbol{\theta}_\star$ is a fixed, unknown vector of real coefficients, and $z_t$ denotes a zero-mean random variable. Traditional linear bandit algorithms utilize regularized least squares (RLS) to estimate the unknown weight $\boldsymbol{\theta}_\star$ as

$$\boldsymbol{A}^{(t)} = \lambda \boldsymbol{I}_d + \left(\boldsymbol{X}^{(t)}\right)^\top \boldsymbol{X}^{(t)} \quad and \quad \hat{\boldsymbol{\theta}}_t = \left(\boldsymbol{A}^{(t)}\right)^{-1} \sum_{s=1}^t r_s \boldsymbol{x}_s, \tag{2}$$

where $\left(\boldsymbol{X}^{(t)}\right)^\top = \left[\boldsymbol{x}_1^\top, \ldots, \boldsymbol{x}_t^\top\right]$ is the $d \times t$ matrix containing all the arms selected up to round $t$ and $\lambda$ is the regularization parameter.

Sketch-based linear bandit methods create a smaller matrix (termed sketch matrix) $\boldsymbol{S}^{(t)} \in \mathbb{R}^{l \times d}$ as an approximation to $\boldsymbol{X}^{(t)}$, where $l$ is the sketch size. Take FD as an example, and we can formulate this sketching operation as

$$\boldsymbol{S}^{(t)} = \sqrt{\left(\boldsymbol{\Sigma}_l^{(t-1)}\right)^2 - \left(\sigma_l^{(t-1)}\right)^2 \boldsymbol{I}_l} \cdot \left(\boldsymbol{V}_l^{(t-1)}\right)^\top, \quad \boldsymbol{M}^{(t)} = \left(\boldsymbol{S}^{(t)} \left(\boldsymbol{S}^{(t)}\right)^\top + \lambda \boldsymbol{I}_l\right)^{-1}, \tag{3}$$

where $\boldsymbol{\Sigma}_l^{(t-1)}, \boldsymbol{V}_l^{(t-1)}$ are the result of rank-$l$ SVD on round $t-1$ and $\boldsymbol{M}^{(t)}$ is a diagonal matrix which can be stored efficiently. According to Woodbury's identity, we can rewrite the inverse of the sketched covariance matrix as

$$\left(\hat{\boldsymbol{A}}^{(t)}\right)^{-1} = \left(\lambda \boldsymbol{I}_d + \left(\boldsymbol{S}^{(t)}\right)^\top \boldsymbol{S}^{(t)}\right)^{-1} = \frac{1}{\lambda}\left(\boldsymbol{I}_d - \left(\boldsymbol{S}^{(t)}\right)^\top \boldsymbol{M}^{(t)} \boldsymbol{S}^{(t)}\right). \tag{4}$$

The computation in equation 2 requires $\Omega\left(d^2\right)$ time. To improve efficiency, sketch-based methods replace $\left(\boldsymbol{A}^{(t)}\right)^{-1}$ with $\left(\hat{\boldsymbol{A}}^{(t)}\right)^{-1}$. Notably, $\left(\hat{\boldsymbol{A}}^{(t)}\right)^{-1}$ can be updated implicitly using the sketch matrix $\boldsymbol{S}^{(t)}$ and $\boldsymbol{M}^{(t)}$ in equation 4. Since matrix-vector multiplications with $\boldsymbol{S}^{(t)}$ take $O(ld)$ time and matrix-matrix multiplications with $\boldsymbol{M}^{(t)}$ take $O\left(l^2\right)$ time, the computation involving the inverse of the sketched covariance matrix is accelerated from $\Omega\left(d^2\right)$ to $O\left(ld + l^2\right)$.

## 2.2 THE MOTIVATION OF REVISITING

Without loss of generality, our analysis below is based on the original FD. To better illustrate our motivation, we first present the complete regret bound for the linear bandit using FD (Kuzborskij et al., 2019).

Let $\overline{\sigma} = \sum_{t=1}^{T} \left( \sigma_l^{(t)} \right)^2$ denote the sum of singular values reduced by FD up to the $T$-th round, where $l$ is the sketch size and $\left( \sigma_l^{(t)} \right)^2$ is the shrinking value at round $t$. According to Liberty (2013), it can be bounded by the *spectral error* $\Delta_T$, i.e., $\overline{\sigma} \le \Delta_T$, where

$$\Delta_T := \min_{k<l} \frac{\left\| \boldsymbol{X}^{(T)} - \boldsymbol{X}_{[k]}^{(T)} \right\|_F^2}{l-k}. \tag{5}$$

Consequently, the regret of the sketch-based linear bandit can be formulated as

$$\text{Regret}_T = \tilde{O}\left( (1+\Delta_T)^{\frac{3}{2}} (l + d\log(1+\Delta_T))\sqrt{T} \right). \tag{6}$$

Denote $k_\star$ as the minimizer of equation 5. Ignoring logarithmic terms, we assume $\Delta_T = T^\gamma$. When $\gamma > \frac{1}{3}$, the regret will exceed $O(T)$. More precisely, when the spectral tail $\left\| \boldsymbol{X}^{(T)} - \boldsymbol{X}_{[k_\star]}^{(T)} \right\|_F^2 = \Omega\left( (l - k_\star)^{\frac{2}{3}} T^{\frac{1}{3}} \right)$, the invalid linear regret will emerge.

The above analysis highlights a key pitfall of sketch-based linear bandits: the pre-set sketch size is crucial. Next, we will explain, from both theoretical and experimental perspectives, why this pitfall is widespread and difficult to avoid in current methods. We present the following theorem to show that the spectral tail of the covariance matrix does not decrease rapidly, even in a non-adversarial setting.

**Theorem 1.** *Suppose the chosen arm $\boldsymbol{x}_t \in \mathbb{R}^d$ at round $t$ is a random vector drawn iid from any distribution over $r \le d$ orthonormal vectors $\boldsymbol{A}$. For any sketch size $l \le r$, the bound on the expected regret of linear bandits using FD is $\Omega\left(T^2\right)$.*

The detailed proof is provided in Appendix A. Theorem 1 shows that when the sketch size is insufficient to capture most of the spectral information, sketch-based linear bandit methods will suffer from linear regret. Furthermore, as illustrated in Figure 1, we observe that an incorrect selection of the pre-set sketch size can significantly degrade performance.

Current methods use the single-scale sketching technique to approximate the covariance matrix. However, since the covariance matrix is determined in an online manner, its spectral information is unknown beforehand. Even if poor performance is detected during learning, adjusting the sketch size is not possible because the shrinking process in single-scale sketching is irreversible. Consequently, the pitfall of linear regret is difficult to avoid in previous works.

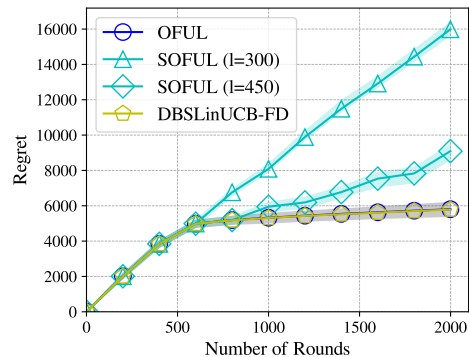

Figure 1: Regret of SOFUL, OFUL, and our method on synthetic data (details in section 5.2) The regret of SOFUL is nearly linear when sketch size $l = 300$.

## 3 DYADIC BLOCK SKETCHING FOR CONSTRAINED GLOBAL ERROR BOUND

In this section, we propose a multi-scale sketching method called Dyadic Block Sketching, which provides a constrained global error bound for matrix sketching. This scalable method converts any streaming sketch into a matrix sketch with a constrained global error bound. It can adaptively adjust the sketch size in a streaming environment, tracking the best rank-$k$ approximation of the target matrix at a minimal cost. We employ the Frequent Directions sketch to illustrate this method.

## 3.1 DECOMPOSABILITY

Dyadic Block Sketching leverages the decomposability of streaming matrix sketches. Intuitively, decomposability means that if the rows of a matrix are divided into submatrices, sketches can be constructed for each submatrix, and these sketches can then be combined to form a sketch of the original matrix. In this process, the error of approximating the original matrix is the sum of the errors of approximating each submatrix. The following lemma demonstrates that all matrix sketches offering a covariance error guarantee exhibit decomposability.

**Lemma 1** (Decomposability). *Given a matrix $\boldsymbol{X} \in \mathbb{R}^{n \times d}$, we decompose $\boldsymbol{X}$ into $p$ submatrices, i.e., $\boldsymbol{X}^\top = \begin{bmatrix} \boldsymbol{X}_1^\top, \boldsymbol{X}_2^\top, ..., \boldsymbol{X}_p^\top \end{bmatrix}$. For any $i \in [p]$, if we construct a matrix sketch with covariance error guarantee $\epsilon_i$ for each sub-matrix $\boldsymbol{X}_i$, denoted as $\boldsymbol{S}_i$, such that $\left\| \boldsymbol{X}_i^\top \boldsymbol{X}_i - \boldsymbol{S}_i^\top \boldsymbol{S}_i \right\|_2 \leq \epsilon_i \cdot \left\| \boldsymbol{X}_i - \boldsymbol{X}_{i[k]} \right\|_F^2$. Then $\boldsymbol{S}^\top = \begin{bmatrix} \boldsymbol{S}_1^\top, \boldsymbol{S}_2^\top, ..., \boldsymbol{S}_p^\top \end{bmatrix}$ is an approximation of $\boldsymbol{X}^\top$ and the error bound is*

$$\left\| \boldsymbol{X}^\top \boldsymbol{X} - \boldsymbol{S}^\top \boldsymbol{S} \right\|_2 \leq \sum_{i=1}^{p} \epsilon_i \cdot \left\| \boldsymbol{X}_i - \boldsymbol{X}_{i[k]} \right\|_F^2 .$$

*Proof.* Since we have $\boldsymbol{X}^\top \boldsymbol{X} = \sum_{i=1}^{p} \boldsymbol{X}_i^\top \boldsymbol{X}_i$ and $\boldsymbol{S}^\top \boldsymbol{S} = \sum_{i=1}^{p} \boldsymbol{S}_i^\top \boldsymbol{S}_i$. Therefore

$$\left\| \boldsymbol{X}^\top \boldsymbol{X} - \boldsymbol{S}^\top \boldsymbol{S} \right\|_2 \leq \sum_{i=1}^{p} \left\| \boldsymbol{X}_i^\top \boldsymbol{X}_i - \boldsymbol{S}_i^\top \boldsymbol{S}_i \right\|_2 \leq \sum_{i=1}^{p} \epsilon_i \cdot \left\| \boldsymbol{X}_i - \boldsymbol{X}_{i[k]} \right\|_F^2 ,$$

and the Lemma follows. $\qquad\square$

## 3.2 ALGORITHM DESCRIPTIONS

**High-Level Ideas.** The high-level idea is illustrated in Figure 2. We establish a logarithmic number of sketch sizes, each partitioning the stream into blocks. The sketch size of each subsequent block is double that of the previous one, thereby halving the maximum error caused by sketching. By maintaining a streaming sketch for each block, we can concatenate all the sketches to approximate the streaming matrix, with the error bounded by the decomposability property.

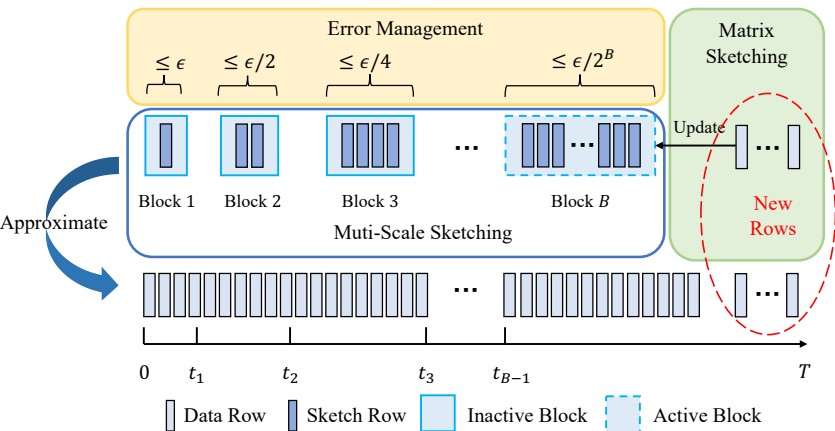

Figure 2: An illustration for Dyadic Block Sketching. For inactive Block $i \in [B - 1]$, the matrix sketch covers the data from $t_{i-1}$ to $t_i$. For the active Block $B$, matrix sketching updates are performed on the new rows. We then merge the multi-scale matrix sketches to approximate the entire stream.

**Algorithm.** We start by defining the data structure for our algorithm. The matrix rows in the stream are divided into blocks, each covering a segment of consecutive, non-overlapping rows. The list of blocks is denoted as $\mathcal{B}$. For $i = 1, 2, \ldots$, each block $\mathcal{B}[i]$ is associated with a streaming sketch of size $l$ (length) and block size (size). The size of block $\mathcal{B}[i]$ is defined as the sum of the squared norms of the rows covered by $\mathcal{B}[i]$, specifically, $\mathcal{B}[i].size = \sum_{\boldsymbol{x} \in \mathcal{B}[i]} \| \boldsymbol{x} \|_2^2$.

We further define two states for the blocks: active and inactive. An active block receives updates, while an inactive block remains entirely fixed. As illustrated in Figure 2, there is always exactly one active block in the stream. Additionally, three key invariants must be maintained:

1. For each inactive block, the length must be greater than the rank of the rows it contains.

2. The sum of sketch rows stored in blocks should be less than $d$.

3. The size of block $\mathcal{B}[i]$ should less than $l_0\epsilon$, where $l_0$ represents the initial sketch size and $\epsilon$ is the error parameter.

Algorithm 1 presents the pseudo-code of Dyadic Block Sketching. When a new row $\boldsymbol{x}_t$ is received, we first verify the maintenance of Invariant 2 (see Line 5). If the block count reaches its upper limit, the error from the streaming sketch becomes intolerable, necessitating the full preservation of the streaming rows' information. Therefore, we execute a complete rank-1 update on the sketch matrix.

---

**Algorithm 1:** Dyadic Block Sketching

**Input:** Data stream $\{\boldsymbol{x}_t\}_{t=1}^T$, sketch size $l_0$, error parameter $\epsilon$, regularization parameters $\lambda$, method $\mathrm{Sk} \in \{\mathrm{FD}, \mathrm{RFD}\}$

**Output:** Sketch matrix $\boldsymbol{S}^{(t)}$, $\boldsymbol{M}^{(t)}$

**1** Initialize $\mathcal{B}[0].size = 0$, $\mathcal{B}[0].length = l_0$, $B = 0$
**2** Initialize $\mathcal{B}[0].sketch$ by method Sk
**3 for** $t \leftarrow 1, \ldots, T$ **do**
**4**      Receive $\boldsymbol{x}_t$
**5**      **if** $B \geq \lfloor \log\left(d/l_0 + 1\right) \rfloor - 1$ **then**
**6**          Update $\left(\boldsymbol{S}^{(t)}\right)^\top \boldsymbol{S}^{(t)} = \left(\boldsymbol{S}^{(t-1)}\right)^\top \boldsymbol{S}^{(t-1)} + \boldsymbol{x}_t^\top \boldsymbol{x}_t$ and $\boldsymbol{M}^{(t)}$ using rank-1
         modifications
**7**      **else**
**8**          **if** $\mathcal{B}[B].size + \|\boldsymbol{x}_t\|^2 > \epsilon \cdot \mathcal{B}[0].length$ *and* $\mathcal{B}[B].length < rank$ **then**
**9**              Initialize $\mathcal{B}[B+1].size = 0$, $\mathcal{B}[B+1].length = 2 \times \mathcal{B}[B].length$
**10**             Initialize $\mathcal{B}[B+1].sketch$ by method Sk
**11**             Set $B = B + 1$
**12**          Update $\mathcal{B}[B].sketch$ with $\boldsymbol{x}_t$
**13**          Set $\boldsymbol{S}_B, \boldsymbol{M}_B, rank \leftarrow \mathcal{B}[B].sketch$
**14**          Update $\mathcal{B}[B].size \mathrel{+}= \|\boldsymbol{x}_t\|^2$
**15**          Initialize empty matrix $\boldsymbol{S}^{(t)}, \boldsymbol{M}^{(t)}$
**16**          **for** $i \leftarrow 0, \ldots, B$ **do**
**17**             Set $\boldsymbol{S}_i, \boldsymbol{M}_i \leftarrow \mathcal{B}[i].sketch$
**18**             Combine $\boldsymbol{S}^{(t)}$ with $\boldsymbol{S}_i$, $\boldsymbol{M}^{(t)}$ with $\boldsymbol{M}_i$ as equation 7

---

In Lines 8 – 11, we control the errors to ensure the maintenance of Invariant 3. If the size of the active block exceeds the specified limit, we store the current block's information and create a new block with double the previous length to prevent further errors.

In Lines 12 – 14, we update the active block's information with $\boldsymbol{x}_t$. During this process, we can query the sketch matrices $\boldsymbol{S}_B$ and $\boldsymbol{M}_B$ in the active block (details in Appendix D). Additionally, the shrinkage of the deterministic sketch provides us with the block's current rank if the sketch size exceeds the block's rank. We use the variable $rank$ to track this value. If the shrinking value is non-zero, we set $rank$ to $\mathcal{B}[B].length$; otherwise, we assign $rank$ to the block's rank.

In Lines 15 – 18, we query the sketch of the entire stream. To retrieve the sketch matrix $\boldsymbol{S}^{(t)}$ and $\boldsymbol{M}^{(t)}$, we combine them with the previous matrices as follows

$$\boldsymbol{S}^{(t)} = \begin{pmatrix} \boldsymbol{S}^{(t)} \\ \boldsymbol{S}_i \end{pmatrix} \quad , \quad \boldsymbol{M}^{(t)} = \left( \begin{pmatrix} \boldsymbol{M}^{(t)} & \boldsymbol{S}^{(t)}(\boldsymbol{S}_i)^\top \\ \boldsymbol{S}_i(\boldsymbol{S}^{(t)})^\top & \boldsymbol{M}_i \end{pmatrix} + \lambda \boldsymbol{I} \right)^{-1}. \tag{7}$$

Since the inactive blocks remain fixed, we can store the combined result of the sketch matrix in the inactive blocks. In practice, we can avoid the looped calculation of equation 7 and perform the combination only with the active block.

### 3.3 ANALYSIS

We explore the error guarantees along with the space and time complexities of the Dyadic Block Sketching method. Initially, we prove a general theorem that establishes the relationship between the complexities of the Dyadic Block Sketching algorithm and the streaming sketch utilized in each block. Subsequently, we examine how a specific deterministic streaming sketch can be integrated into the Dyadic Block Sketching framework. We present a theorem detailing the space usage and update costs associated with Dyadic Block Sketching. The detailed proof is provided in Appendix B.

**Theorem 2.** *Suppose a streaming matrix sketch, denoted as $\kappa$, achieves a covariance error $\left\| \boldsymbol{X}^\top \boldsymbol{X} - \boldsymbol{S}^\top \boldsymbol{S} \right\|_2 \leq \eta \cdot \left\| \boldsymbol{X} \right\|_F^2$ with $\ell_\eta$ rows and $\mu_\eta$ update time. Applying $\kappa$ as the sketching method for each block in the Dyadic Block Sketching and $l_0$ is the initial sketch size, we generate a matrix sketch $\boldsymbol{S}$ for the entire streaming matrix $\boldsymbol{X}$ with an error guarantee $\left\| \boldsymbol{X}^\top \boldsymbol{X} - \boldsymbol{S}^\top \boldsymbol{S} \right\|_2 \leq 2\epsilon$. Assuming $\mathrm{rank}(\boldsymbol{X}) = k$ and that the rows are normalized, the space cost for Dyadic Block Sketching is $O\left( d \cdot \sum_{i=0}^B \ell_{\frac{1}{2^i l_0}} \right)$, and the update cost is $O\left( \mu_{\frac{1}{2^B l_0}} \right)$, where $B = \left\lceil \min\left\{ \log \frac{k}{l_0}, \frac{T}{\epsilon l_0} \right\} \right\rceil + 1$.*

Note that different streaming sketches will result in varying costs. To illustrate this, we provide a corollary using the well-known deterministic sketching method Frequent Directions (FD).

**Dyadic Block Sketching for FD.** This algorithm employs the FD sketch for each block in the Dyadic Block Sketching framework. Recall that with a given error parameter $\eta$, the FD sketch requires a space of $\ell_\eta = O(1/\eta)$ and processes updates at an amortized cost of $\mu_\eta = O(d/\eta)$. As outlined in Theorem 2, we derive the following corollary:

**Corollary 1.** *The Dyadic Block Sketching algorithm for FD uses $O\left(2dk - dl_0\right)$ space and processes an update with $O(dk)$ amortized cost.*

**Remark 1** (Efficient Implementation). *The primary computational costs of the algorithm include calculating the SVD to obtain $\boldsymbol{S}^{(t)}$ and performing matrix multiplication to compute $\boldsymbol{M}^{(t)}$, both of which cost $O(dl^2)$, where $l$ is the current sketch size. However, the amortized update cost can be effectively reduced from $O(dl^2)$ to $O(dl)$ either by doubling the space, as detailed in Algorithm 3 in Appendix C, or by employing the Gu-Eisenstat procedure (Gu & Eisenstat, 1993).*

**Remark 2** (Worst-Case Analysis). *Compared to the single streaming sketch algorithm, our method effectively controls the global error of matrix approximation by limiting error sizes within each block, thus enabling dynamic adjustment of the sketch size. Particularly when dealing with a full-rank matrix with a heavy spectral tail, sketching methods should be avoided to prevent the pitfall of linear regret. However, this control is impossible with a single sketch due to the irreversible nature of the shrinking process in sketching. In the worst-case scenario, our method ensures that the streaming matrix problem can be dynamically adjusted to revert to a non-sketch situation.*

## 4 LINEAR BANDITS THROUGH DYADIC BLOCK SKETCHING

In this section, we introduce a novel framework for efficient sketch-based linear bandits, termed DBSLinUCB, which leverages Dyadic Block Sketching. As outlined in Section 2.2, a key limitation of previous methods stems from their reliance on single-scale matrix sketching, resulting in a *space-bounded* linear bandit approach. The use of a fixed sketch size leads to uncontrollable spectral loss, $\Delta_T$, ultimately causing linear regret.

In contrast, Dyadic Block Sketching employs a multi-scale matrix sketching strategy, where the sketch size is adaptively adjusted based on the pre-set parameter $\epsilon$. Consequently, DBSLinUCB is an *error-bounded* linear bandit method that effectively addresses and overcomes the pitfalls of linear regret present in existing approaches.

The procedure, detailed in Algorithm 2, builds on prior sketch-based algorithms but incorporates the Dyadic Block Sketching method to effectively manage the error in approximating the covariance

---

**Algorithm 2:** DBSLinUCB

---

**Input:** Data stream $\{\boldsymbol{x}_t\}_{t=1}^T$, sketch size $l_0$, error parameter $\epsilon$, regularization parameters $\lambda$,
method Sk $\in \{\text{FD}, \text{RFD}\}$, confidence $\delta$

**1** Initialize a Dyadic Block Sketching instance $sketch\left(\boldsymbol{S}^{(0)}, \boldsymbol{M}^{(0)}\right)$ with $l_0, \lambda$, method Sk

**2 for** $t \leftarrow 1, \ldots, T$ **do**

**3**     Get arm set $\mathcal{X}_t$

**4**     Compute the confidence ellipsoid $\hat{\beta}_{t-1}(\delta)$ using equation 9

**5**     Select $\boldsymbol{x}_t = \underset{\boldsymbol{x} \in \mathcal{X}_t}{\arg\max} \left\{ \boldsymbol{x}^\top \hat{\boldsymbol{\theta}}_{t-1} + \hat{\beta}_{t-1}(\delta) \cdot \|\boldsymbol{x}\|_{\left(\hat{\boldsymbol{A}}^{(t-1)}\right)^{-1}} \right\}$

**6**     Receive the reward $r_t$

**7**     Update $sketch\left(\boldsymbol{S}^{(t)}, \boldsymbol{M}^{(t)}\right)$ with $\boldsymbol{x}_t$ and compute $\hat{\boldsymbol{A}}^{(t)}$ and $\hat{\boldsymbol{\theta}}_t$ using equation 8

---

matrix. At each round $t$, we employ the sketch matrix $\boldsymbol{S}^{(t)}$ to approximate the covariance matrix, from which we derive the sketched regularized least squares estimator as follows

$$\hat{\boldsymbol{A}}^{(t)} = \boldsymbol{S}^{(t)^\top} \boldsymbol{S}^{(t)} + \lambda \boldsymbol{I} \quad , \quad \hat{\boldsymbol{\theta}}_t = \left(\hat{\boldsymbol{A}}^{(t)}\right)^{-1} \sum_{s=1}^t r_s \boldsymbol{x}_s. \tag{8}$$

Denote $\mathcal{X}_t$ as the set of arms available at round $t$ and $\hat{\beta}_{t-1}(\delta)$ as the confidence ellipsoid, $l_{B_t}$ as the sketch size in the active block. The arms selected by DBSLinUCB are determined by solving the following constrained optimization problem:

$$\boldsymbol{x}_t = \underset{\boldsymbol{x} \in \mathcal{X}_t}{\arg\max} \max_{\boldsymbol{\theta} \in \mathbb{R}^d} \boldsymbol{x}^\top \boldsymbol{\theta} \quad \text{such that} \quad \left\|\boldsymbol{\theta} - \hat{\boldsymbol{\theta}}\right\|_{\hat{\boldsymbol{A}}^{(t-1)}} \leq \hat{\beta}_{t-1}(\delta). \tag{9}$$

The updates to the sketched regularized least squares estimator and the calculations for the confidence ellipsoid can be efficiently completed in $O\left(dl_{B_t} + l_{B_t}^2\right)$ time using the Woodbury identity as stated in equation 4. This makes DBSLinUCB significantly more efficient than traditional linear bandit algorithms, which require $\Omega\left(d^2\right)$ in both time and space.

DBSLinUCB represents a scalable framework for efficient sketch-based linear bandits that are capable of incorporating various streaming sketching techniques. We now explore two deterministic sketching techniques that provide different regret bounds of linear bandits.

**DBSLinUCB using FD.**    We explore the Frequent Directions (FD) (see Algorithm 4), a deterministic sketching method (Liberty, 2013; Ghashami et al., 2016). FD uniquely maintains the invariant that the last row of the sketch matrix, $\boldsymbol{S}$, is always zero. In each round, a new row $\boldsymbol{a}_t$ is inserted into this last row of $\boldsymbol{S}$, and the matrix undergoes singular value decomposition into $\boldsymbol{U}\boldsymbol{\Sigma}\boldsymbol{V}^\top$. Subsequently, $\boldsymbol{S}$ is updated to $\sqrt{\boldsymbol{\Sigma}_l^2 - \sigma \boldsymbol{I}} \cdot \boldsymbol{V}_l^\top$, where $\sigma$ represents the square of the $l$-th singular value. Given that the rows of $\boldsymbol{S}$ are orthogonal, $\boldsymbol{M} = (\boldsymbol{S}\boldsymbol{S}^\top + \lambda \boldsymbol{I})^{-1}$ remains a diagonal matrix, facilitating efficient maintenance. We integrate FD into DBSLinUCB and established the following regret bound:

**Theorem 3.** *Assume that* $\|\boldsymbol{\theta}_\star\|_2 \leq H$, $\|\boldsymbol{x}\|_2 \leq L$, *and* $L \geq \sqrt{\lambda}$. *Suppose that the noise is conditionally $R$-subgaussian, where $R$ is a fixed constant. The sketch size in the active block at round $t$ is denoted as $l_{B_t}$. Given the error parameter $\epsilon$, then with a probability of $1 - \frac{1}{T}$, the regret of Algorithm 2 utilizing* Sk = FD *is*

$$Regret_T \overset{\tilde{O}}{=} \frac{L(R + H\sqrt{\lambda})}{\sqrt{\lambda}} \cdot \left(d \ln\left(1 + \frac{\epsilon}{\lambda}\right) + 2l_{B_T} \cdot \ln\left(1 + \frac{TL^2}{2l_{B_T}\lambda}\right)\right) \cdot \left(1 + \frac{\epsilon}{\lambda}\right)^{\frac{3}{2}} \sqrt{T}.$$

**Remark 3.** *Compared to linear bandits that use a single FD sketch, our approach relies on a predetermined error parameter $\epsilon$ rather than fixed sketch size $l$. This ensures a sublinear regret bound of order $\tilde{O}(\sqrt{T})$ without requiring prior knowledge of the streaming matrix. The sketch size in DBSLinUCB can be dynamically adjusted to match the desired order of regret. In practice, if the goal is to significantly enhance efficiency at the expense of a higher order of regret, $\epsilon$ can be set as a function of $T$. Another advantage of DBSLinUCB is that the initial sketch size is independent of the error term $\epsilon$; it only affects the total computational and space cost.*

**DBSLinUCB using RFD.** We employ the Robust Frequent Directions (RFD) sketch (see Algorithm 5), a sketching strategy designed to address the rank deficiency issue inherent in FD (Luo et al., 2019). RFD reduces the approximation error of FD by maintaining a counter $\alpha$, which quantifies the spectral error. More precisely, RFD employs $\boldsymbol{S}^\top\boldsymbol{S} + \alpha\boldsymbol{I}$ to approximate $\boldsymbol{A}^\top\boldsymbol{A}$. We integrate RFD into DBSLinUCB and established the following regret bound:

**Theorem 4.** *Assume that $\|\boldsymbol{\theta}_\star\|_2 \leq H$, $\|\boldsymbol{x}\|_2 \leq L$, and $L \geq \sqrt{\lambda}$. Suppose that the noise is conditionally R-subgaussian, where $R$ is a fixed constant. The sketch size in the active block at round $t$ is denoted as $l_{B_t}$. Given the error parameter $\epsilon$, then with a probability of $1 - \frac{1}{T}$, the regret of Algorithm 2 utilizing $\mathrm{Sk} = \mathrm{RFD}$ is*

$$Regret_T \stackrel{\tilde{\mathcal{O}}}{=} \frac{L}{\sqrt{\lambda}} \cdot \sqrt{\ln\left(1 + \frac{TL^2}{2l_{B_T}\lambda} + \left(1 - \frac{2 - B_T}{2^{B_T+1}}\right) \cdot \frac{\epsilon}{\lambda}\right)} \cdot \left(H \cdot \sqrt{\lambda + \epsilon} + \right.$$

$$\left. R \cdot \sqrt{d\ln\left(1 + \frac{\epsilon}{\lambda}\right) + 2l_{B_T} \cdot \ln\left(1 + \frac{TL^2}{2l_{B_T}\lambda} + \left(1 - \frac{2 - B_T}{2^{B_T+1}}\right) \cdot \frac{\epsilon}{\lambda}\right)}\right).$$

**Remark 4.** *Compared to DBSLinUCB using FD, the order of the error term $\epsilon$ is reduced from $3/2$ to $1/2$. Apart from logarithmic terms, decoupling the dimensions $d$ and $\epsilon$ further reduces the impact of the error. These properties indicate that Dyadic Block Sketching maintains the excellent characteristics of RFD. In fact, despite having the same error bound as DBSLinUCB using FD, DBSLinUCB using RFD satisfies both positive definite monotonicity and well-conditioned properties. We present details and proof in Appendix G.*

## 5 EXPERIMENTS

In this section, we empirically verify the efficiency and effectiveness of our algorithms. We conduct experiments on both synthetic and real-world datasets. Each experiment is performed over 20 different random permutations of the datasets. All experiments are performed on a machine with 24-core Intel(R) Xeon(R) Gold 6240R 2.40GHz CPU and 256 GB memory.

### 5.1 MATRIX APPROXIMATION

We evaluate the performance of the proposed Dyadic Block Sketching in terms of matrix approximation. We compare it with FD (Liberty, 2013). We generated a synthetic dataset with $n = 1250$ rows and $d = 100$ columns. Specifically, each row $\boldsymbol{a}_t \in \mathbb{R}^{100}$ is independently drawn from a multivariate Gaussian distribution $\boldsymbol{a}_t \sim \mathcal{N}(\boldsymbol{0}, \boldsymbol{I}_d)$. We set the sketch size $l_0 = 50$ for FD. We set the initial sketch size $l_0 = 16$ and the error parameter $\epsilon = 2000$ for Dyadic Block Sketching.

Figure 3 shows the spectral norm error $\|\boldsymbol{A}_t^\top\boldsymbol{A}_t - \boldsymbol{S}_t^\top\boldsymbol{S}_t\|_2$ and its upper bound for matrix sketching, where $\boldsymbol{A}_t$ is the steaming matrix at round $t$ and $\boldsymbol{S}_t$ is the skech matrix at round $t$. We observe that Dyadic Block Sketching provides a constrained global error bound for matrix sketching. Compared with FD, the rate of error growth in Dyadic Block

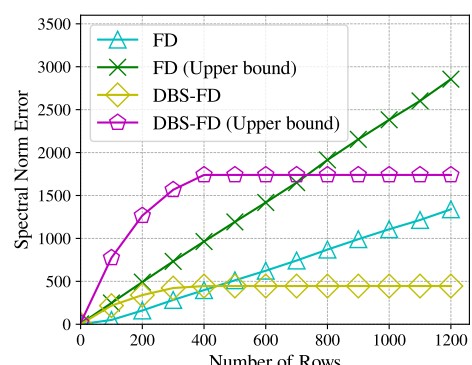

Figure 3: Comparison among FD and our DBS-FD w.r.t. the error and its upper bound

Sketching decreases over time, effectively limiting the linear growth of the spectral tail.

### 5.2 ONLINE REGRESSION IN SYNTHETIC DATA

In this section, we evaluate our DBSLinUCB on synthetic datasets. The baselines include the non-sketched method OFUL (Abbasi-Yadkori et al., 2011) and the sketch-based methods SO-FUL (Kuzborskij et al., 2019), CBSCFD (Chen et al., 2021). Inspired by the experimental settings in Chen et al. (2021), we build synthetic datasets using multivariate Gaussian distributions $\mathcal{N}(0, \boldsymbol{I}_d)$

with 100 arms and $d = 500$ features per context. The true parameter $\theta_\star$ is drawn from $\mathcal{N}(0, \mathbf{I}_d)$ and is normalized. The confidence ellipsoid $\beta$ of all algorithms is searched in $\{10^{-4}, 10^{-3}, \ldots, 1\}$ and $\lambda$ is searched in $\{2 \times 10^{-4}, 2 \times 10^{-3}, \ldots, 2 \times 10^4\}$. We set the sketch size $l = 300, 450$ for SOFUL and CBSCFD and the initial sketch size $l_0 = 64$ for DBSLinUCB. Additionally, we set the error parameter $\epsilon = 2000$ for DBSLinUCB.

Experimental results in Figure 1 (in Section 2.2), 4a show that DBSLinUCB using FD and RFD consistently outperforms the other sketch-based algorithms in terms of the regret of online learning. We observe that when $l = 300$, SOFUL and CBSCFD perform significantly worse than DBSLinUCB, with SOFUL exhibiting nearly linear regret. Moreover, DBSLinUCB achieves sublinear regret similar to OFUL by providing a constrained global error bound. Our experimental results confirm our analysis in Section 2.2, indicating that for all existing sketch-based linear bandit algorithms, inappropriate sketch size selection can lead to the pitfall of linear regret.

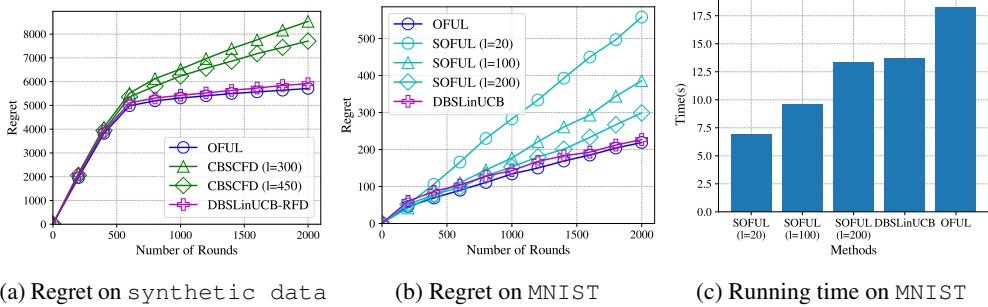

(a) Regret on `synthetic data`     (b) Regret on `MNIST`     (c) Running time on `MNIST`

Figure 4: (a): Cumulative regret of the compared algorithms, the proposed DBSLinUCB using RFD on a synthetic dataset; (b), (c): Cumulative regret and total running time of the compared algorithms, the proposed DBSLinUCB using FD on `MNIST`

### 5.3 ONLINE CLASSIFICATION IN REAL-WORLD DATA

We perform online classification on the real-world dataset `MNIST` to evaluate the performance of our methods. The dataset contains $60,000$ samples, each with $d = 784$ features, and there are $M = 10$ possible labels for each sample. We follow the experimental setup in Kuzborskij et al. (2019). Specifically, we construct the online classification problem within the contextual bandit setting as follows: given a dataset with data in $M$ labels, we first choose one cluster as the target label. In each round, we randomly draw one sample from each label and compose an arm set of $M$ samples in $M$ contexts. The algorithms choose one sample from the arm set and observe the reward based on whether the selected sample belongs to the target label. The reward is $1$ if the selected sample comes from the target label and $0$ otherwise. We set sketch size $l = 20, 100, 200$ for SOFUL and $l_0 = 2$ for DBSLinUCB. We set the error parameter $\epsilon = 1000$ for DBSLinUCB. The choice of confidence ellipsoid and regularization parameter follows the previous section.

Figures 4b and 4c compare the online mistakes and running times of different algorithms. Our findings indicate that, for a given dataset, there exists an optimal sketch size (e.g., $l = 200$) that captures most of the spectral information of the original matrix, thereby accelerating the algorithm without significantly compromising performance. However, selecting this optimal sketch size for SOFUL is challenging due to the lack of prior knowledge about the data. When $l = 20$ or $l = 100$, the regret of SOFUL is significantly worse than that of the non-sketched method, OFUL. In contrast, DBSLinUCB matches the performance of OFUL by adaptively adjusting the sketch size to the near-optimal value of $l = 200$ while being significantly faster than OFUL.

## 6 CONCLUSION

This paper addresses the current pitfall of linear regret in sketch-based linear bandits for the first time. We propose Dyadic Block Sketching with a constrained global error bound and provide formal theoretical guarantees. By leveraging Dyadic Block Sketching, we present a framework for efficient sketch-based linear bandits. Even in the worst-case scenario, our method can achieve sublinear regret without prior knowledge of the covariance matrix. Extensive experimental evaluations on real and synthetic datasets demonstrate the excellent performance and efficiency of our methods.

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

## A    PROOF OF THEOREM 1

For linear bandits using FD, at round $t$, the sketch matrix $\boldsymbol{S}^{(t-1)} \in \mathbb{R}^{l \times d}$ is utilized, and a new row $\boldsymbol{x}_t$ is received, where $l$ denotes the sketch size. Let $\boldsymbol{\Sigma} = \mathbb{E}[\boldsymbol{x}_t \boldsymbol{x}_t^\top]$ be the covariance matrix of the arms, and let $\lambda_i$ be the $i$-th eigenvalue of $\boldsymbol{\Sigma}$. By definition, $\boldsymbol{x}_t$ follows the distribution $\boldsymbol{x}_t = \boldsymbol{a}_i$ with probability $\lambda_i$, where $\boldsymbol{a}_i$ is the $i$-th vector of $\boldsymbol{A}$.

We first consider the expected value of the shrinking factor when the sketch is full-rank. If rank $\left(\boldsymbol{S}^{(t-1)}\right) = l - 1$. Let $\boldsymbol{S}^{(t-1)} = \boldsymbol{U}\boldsymbol{\Sigma}\boldsymbol{V}$ be the SVD of $\boldsymbol{S}^{(t-1)}$, and $\boldsymbol{v}_i$ be the $i$-th row of $\boldsymbol{V}$. Denote the set of basis vectors not in the row space of $\boldsymbol{S}^{(t-1)}$ as $W_{t-1} = \boldsymbol{A} \backslash \{\boldsymbol{v}_1, ..., \boldsymbol{v}_{l-1}\}$, we have $|W_{t-1}| = r - l + 1$. If $\boldsymbol{x}_t \in \text{span}(\boldsymbol{v}_1, ..., \boldsymbol{v}_{l-1})$, we have the shrinking value $\left(\sigma_l^{(t)}\right)^2 = 0$; otherwise $\left(\sigma_l^{(t)}\right)^2 = 1$, with probability $\sum_{a_i \in W_{t-1}} \lambda_i \geq \sum_{i=l}^r \lambda_i$. Therefore, If rank$(\boldsymbol{S}^{(t-1)}) = l - 1$, we have

$$\mathbb{E}\left[\left(\sigma_l^{(t)}\right)^2 \mid \boldsymbol{S}^{(t-1)}\right] \geq \sum_{i=l}^r \lambda_i.$$

If rank $\left(\boldsymbol{S}^{(t-1)}\right) < l - 1$, this means that $\boldsymbol{S}^{(t-1)}$ contains fewer than $l - 1$ distinct vectors drawn from $\boldsymbol{A}$. Let $I_i$ be the indicator variable for drawing $a_i$ in the first $t - 1$ rounds. Then we have the expected number of distinct vectors at round $t - 1$

$$\mathbb{E}\left[\sum_{i=1}^r I_i\right] = \sum_{i=1}^r \left(1 - (1 - \lambda_i)^{t-1}\right).$$

Using Markov's inequality, we have

$$\text{Prob}\left[r - \sum_{i=1}^r I_i \geq r - l + 2\right] \leq \frac{\sum_{i=1}^r 1 - (1 - \lambda_i)^{t-1}}{r - l + 2} \leq \frac{r(1 - \lambda_r)^{t-1}}{2}.$$

Note that this is precisely the probability of having fewer than $l - 1$ distinct vectors in the first $t - 1$ rounds. We conclude that for $t \geq \log\left(\frac{r}{\lambda_r}\right) + 1$, $\text{Prob}\left[\text{rank}(\boldsymbol{S}_{t-1}) = l - 1\right] \geq \frac{1}{2}$. This implies that $\mathbb{E}[\rho_t] \geq \frac{1}{2} \sum_{i=l}^r \lambda_i$ after an initial logarithmic number of rounds.

Therefore, assuming $T \geq 2 \log\left(\frac{r}{\lambda_r}\right)$, the expected accumulated shrinking value is at least:

$$\mathbb{E}\left[\sum_{t=1}^T \left(\sigma_l^{(t)}\right)^2\right] \geq \frac{T}{4} \sum_{i=l}^r \lambda_i.$$

Note that the accumulated shrinking value is upper-bounded by the spectral loss. According to the regret in equation 6, we conclude that the regret upper bound is $\Omega\left(T^2\right)$ in expectation.

## B    PROOF OF THEOREM 2

We begin our proof by considering the number of blocks. Let the entire data stream be denoted as $\boldsymbol{X}^\top = \left[\boldsymbol{X}_0^\top, \boldsymbol{X}_1^\top, \ldots, \boldsymbol{X}_B^\top\right]$. Block $i$ covers the submatrix $\boldsymbol{X}_i$ and stores the corresponding matrix sketch $\boldsymbol{S}_i$.

According to Invariant 3, each submatrix $\boldsymbol{X}_i$ in block $i$ contains at least $\lfloor \epsilon l_0 \rfloor$ rows. Since there are $T$ rows available for allocation in the entire matrix, the maximum number of blocks is $\left\lceil \frac{T}{\epsilon l_0} \right\rceil$.

When $\epsilon$ is small, the block length grows exponentially. By Invariant 1, the length of the last active block will be exactly greater than $k$, i.e., $2^{B-1} l_0 \leq k \leq 2^B l_0$.

Thus, we take the minimum of the two scenarios to determine the number of blocks, resulting in
$B = \left\lceil \min\left\{ \log \frac{k}{l_0}, \frac{T}{\epsilon l_0} \right\} \right\rceil + 1.$

For the error guarantee, recall that the error comprises two components: the error from the active block and the error from merging $B$ inactive blocks. Given that the streaming sketch can detect the best rank-$k$ approximation as long as the sketch size exceeds $k$, the approximation error in the $B$-th active block is effectively zero.

By maintaining Invariant 3, the size of the $i$-th inactive block is bounded by $l_0\epsilon$. Additionally, the $i$-th block employs a streaming matrix sketch with an error parameter of $\frac{1}{2^i l_0}$, thereby ensuring that the maximum error introduced by a sketch at the $i$-th block is at most $\frac{\epsilon}{2^i}$.

Combining all $B + 1$ sketches, we use $\boldsymbol{S}^\top = \left[ \boldsymbol{S}_0^\top, \boldsymbol{S}_1^\top, \ldots, \boldsymbol{S}_B^\top \right]$ to approximate $\boldsymbol{X}$. By Lemma 1, this provides the following error guarantee for the entire streaming matrix:

$$
\begin{aligned}
\left\| \boldsymbol{X}^\top \boldsymbol{X} - \boldsymbol{S}^\top \boldsymbol{S} \right\|_2 &\leq \sum_{i=0}^{B} \left\| \boldsymbol{X}_i^\top \boldsymbol{X}_i - \boldsymbol{S}_i^\top \boldsymbol{S}_i \right\|_2 \\
&= \sum_{i=0}^{B-1} \left\| \boldsymbol{X}_i^\top \boldsymbol{X}_i - \boldsymbol{S}_i^\top \boldsymbol{S}_i \right\|_2 + \left\| \boldsymbol{X}_B^\top \boldsymbol{X}_B - \boldsymbol{S}_B^\top \boldsymbol{S}_B \right\|_2 \\
&\leq \sum_{i=0}^{B-1} \frac{\epsilon}{2^i} + 0 \\
&\leq 2\epsilon.
\end{aligned}
$$

For space usage, the $i$-th block employs a streaming matrix sketch with an error parameter of $\frac{1}{2^i l_0}$, resulting in a sketch of size $\ell_{\frac{1}{2^i l_0}}$. Therefore, the total number of sketched rows is $\sum_{i=0}^{B} \ell_{\frac{1}{2^i l_0}}$. Consequently, the total space requirement is $O\left( d \cdot \sum_{i=0}^{B} \ell_{\frac{1}{2^i l_0}} \right)$.

For the update cost, since only the active sketch requires updating, the cost is $O\left( \mu_{\frac{1}{2^B l_0}} \right)$.

## C  FAST ALGORITHM OF DYADIC BLOCK SKETCHING

The computational cost of FD and RFD, as detailed in Algorithm 4 and 5, is primarily driven by the singular value decomposition (SVD) operations. At round $t$, with $B_t + 1$ blocks, we denote $l_i$ as the sketch size for the $i$-th block. It incurs an amortized time of $O\left( dl_{B_t}^2 \right)$ due to standard SVD processes in the active block. Additionally, the operation to compute $\boldsymbol{M}^{(t)}$ via matrix multiplication and matrix inversion also requires $O\left( \sum_{i=0}^{B_t-1} l_i \cdot l_{B_t} \cdot d + \left( \sum_{i=0}^{B_t} l_i \right)^3 \right) = O\left( dl_{B_t}^2 \right)$. We can enhance the efficiency of our Dyadic Block Sketching by doubling the sketch size, as detailed in Algorithm 3.

Notice that within each epoch, the update of $\boldsymbol{M}^{(t)}$ can be formulate as

$$
\boldsymbol{M}^{(t)} = \begin{pmatrix} \boldsymbol{M}^{(t-1)} + \frac{\phi\phi^\top}{\xi} & \frac{-\phi}{\xi} \\ \frac{-\phi^\top}{\xi} & \frac{1}{\xi} \end{pmatrix}, \tag{10}
$$

where $\phi = \boldsymbol{M}^{(t-1)} \boldsymbol{S}^{(t-1)} \boldsymbol{x}_t^\top$ and $\xi = \boldsymbol{x}_t \boldsymbol{x}_t^\top - \boldsymbol{x}_t \left( \boldsymbol{S}^{(t-1)} \right)^\top \phi + \alpha + \lambda$.

When the method Sk = FD, $\alpha$ is set to 0; conversely, when the method Sk = RFD, $\alpha$ serves as the counter maintained in the RFD sketch.

Given that the length of $\boldsymbol{M}^{(t)}$ is at most twice the length of $\mathcal{B}[B].length$, the amortized computation time required for $\boldsymbol{M}^{(t)}$ is limited to $O\left( dl_{B_t} \right)$. Additionally, we perform the SVD only after every addition of $\mathcal{B}[B].length$ rows, reducing the amortized update time complexity to $O\left( dl_{B_t} \right)$.

---

**Algorithm 3:** Fast Dyadic Block Sketching

**Input:** Data stream $\{x_t\}_{t=1}^T$, sketch size $l_0$, error parameter $\epsilon$, regularization parameters $\lambda$, method $\text{Sk} \in \text{FD}, \text{RFD}$

**Output:** Sketch matrix $S^{(t)}, M^{(t)}$

1 Initialize $\mathcal{B}[0].size = 0, \mathcal{B}[0].length = l_0, B = 0$
2 Initialize $\mathcal{B}[0].sketch$ by method Sk
3 **for** $t \leftarrow 1, \ldots, T$ **do**
4     Receive $x_t$
5     **if** $B \geq \lfloor \log (d/l_0 + 1) \rfloor - 1$ **then**
6        Update $\left(S^{(t)}\right)^\top S^{(t)} = \left(S^{(t-1)}\right)^\top S^{(t-1)} + x_t^\top x_t$
7        Update $M^{(t)}$ using rank-1 modifications
8     **else**
9        **if** $\mathcal{B}[B].size + \|x_t\|^2 > \frac{\epsilon}{2} \cdot \mathcal{B}[0].length$ *and* $\mathcal{B}[B].length < rank$ **then**
10           Initialize $\mathcal{B}[B+1].size = 0, \mathcal{B}[B+1].length = 2 \times \mathcal{B}[B].length$
11           Initialize $\mathcal{B}[B+1].sketch$ by method Sk
12           Set $B \leftarrow B + 1$
13        Append $x_t$ below $\mathcal{B}[B].sketch$
14        Update $\mathcal{B}[B].size += \|x_t\|^2$
15        Initialize empty matrix $S, M$
16        **for** $i \leftarrow 0, \ldots, B - 1$ **do**
17           Set $S_i, M_i \leftarrow \mathcal{B}[i].sketch$
18           Combine $S$ with $S_i$
19           Combine $M$ with $M_i$
20        **if** $\mathcal{B}[B].sketch$ have $2 \cdot \mathcal{B}[B].length$ rows **then**
21           Update $\mathcal{B}[B].sketch$ by method Sk
22           Set $S_B, M_B, rank \leftarrow \mathcal{B}[B].sketch$
23           Update $S^{(t)} = \begin{pmatrix} S \\ S_B \end{pmatrix}, M^{(t)} = \left( \begin{pmatrix} M & S(S_B)^\top \\ S_B S^\top & M_B \end{pmatrix} + \lambda I \right)^{-1}$
24        **else**
25           Update $S^{(t)} = \begin{pmatrix} S \\ S_B \end{pmatrix}$
26           Update $M^{(t)}$ by equation 10

---

## D   PSEUDO-CODE OF DETERMINISTIC MATRIX SKETCHING

---

**Algorithm 4:** FD sketch

**Input:** Data $X \in \mathbb{R}^{T \times d}$, sketch size $l$, regularization $\lambda$

**Output:** Sketch $S, M$

1 Initialize $S \leftarrow \mathbf{0}^{l \times d}, M \leftarrow \frac{1}{\lambda} I_l$
2 **for** $t \leftarrow 1, \ldots, T$ **do**
3     Append $x_t$ to the last row of $S$
4     Compute $[U, \Sigma, V] \leftarrow \text{svd}(S)$
5     Set $\sigma \leftarrow \sigma_l^2$
6     Update $S \leftarrow \sqrt{\Sigma_l^2 - \sigma I} \cdot V_l^\top$
7     Update
       $M \leftarrow \text{diag}\left\{ \frac{1}{\lambda + \sigma_1^2 - \sigma}, \ldots, \frac{1}{\lambda} \right\}$

---

**Algorithm 5:** RFD sketch

**Input:** Data $X \in \mathbb{R}^{T \times d}$, sketch size $l$, regularization $\lambda$

**Output:** Sketch $S, M$ and counter $\alpha$

1 Initialize $S \leftarrow \mathbf{0}^{l \times d}, M \leftarrow \frac{1}{\lambda} I_l, \alpha \leftarrow 0$
2 **for** $t \leftarrow 1, \ldots, T$ **do**
3     Append $x_t$ to the last row of $S$
4     Compute $[U, \Sigma, V] \leftarrow \text{svd}(S)$
5     Set $\sigma \leftarrow \sigma_l^2, \alpha \leftarrow \alpha + \sigma$
6     Update $S \leftarrow \sqrt{\Sigma_l^2 - \sigma I} \cdot V_l^\top$
7     Set
       $M \leftarrow \text{diag}\left\{ \frac{1}{\lambda + \sigma_1^2 - \sigma + \alpha}, \ldots, \frac{1}{\lambda + \alpha} \right\}$

---

The pseudo-code for deterministic matrix sketching methods is displayed in Algorithm 4 and Algorithm 5. Note that these deterministic matrix sketching methods can be accelerated by doubling the sketch size. More details can be found in Appendix C.

## E    PROOF OF THEOREM 3

Following Abbasi-Yadkori et al. (2011), we structure the proof in two parts: first, we establish bounds for the approximate confidence ellipsoid, and second, we delineate the bounds for regret. Denote $B_t$ as the number of blocks at round $t$, and $\overline{\sigma}_i$ as the sum of shrinking singular values in the sketch of block $i$. Let $l_{B_t}$ be the sketch size in the active block at round $t$. We start by establishing an intermediate result concerning the confidence ellipsoid.

**Theorem 5.** *Let $\hat{\boldsymbol{\theta}}_t$ be the RLS estimate constructed by an arbitrary policy for linear bandits after $t$ rounds of play. For any $\delta \in (0,1)$, the optimal unknown weight $\boldsymbol{\theta}_\star$ belongs to the set $\Theta_t \equiv \left\{ \boldsymbol{\theta} \in \mathbb{R}^d : \left\| \boldsymbol{\theta} - \hat{\boldsymbol{\theta}}_t \right\|_{\hat{\boldsymbol{A}}^{(t)}} \leq \hat{\beta}_t(\delta) \right\}$ with probability at least $1 - \delta$, where*

$$
\hat{\beta}_t(\delta) = R \cdot \sqrt{1 + \frac{\sum_{i=1}^{B_t} \overline{\sigma}_i}{\lambda}} \cdot \sqrt{2 \ln\left(\frac{1}{\delta}\right) + d \ln\left(1 + \frac{\sum_{i=1}^{B_t} \overline{\sigma}_i}{\lambda}\right) + 2 l_{B_t} \cdot \ln\left(1 + \frac{t L^2}{2 l_{B_t} \lambda}\right)}
$$

$$
+ H \cdot \frac{\lambda + \sum_{i=1}^{B_t} \overline{\sigma}_i}{\sqrt{\lambda}}.
$$

*Proof.* According to Algorithm 2, the approximate covariance matrix is

$$
\hat{\boldsymbol{A}}^{(t)} = \lambda \boldsymbol{I} + \sum_{i=1}^{B_t} \left( \boldsymbol{S}_i^{(t)} \right)^\top \boldsymbol{S}_i^{(t)},
$$

where $\boldsymbol{S}_i^{(t)}$ is the sketch matrix in block $i$ at round $t$. Define $\boldsymbol{z}_1^\top, ..., \boldsymbol{z}_t^\top \in \mathbb{R}^d$ is the noise sequence conditionally R-subgaussian for a fixed constant $R$ and $\boldsymbol{r}_t^\top = (r_1, r_2, ... r_t) \in \mathbb{R}^d$ is the reward vector. We begin by noticing that

$$
\hat{\boldsymbol{\theta}}_t = \left( \hat{\boldsymbol{A}}^{(t)} \right)^{-1} \boldsymbol{X}_t^\top \boldsymbol{r}_t = \left( \hat{\boldsymbol{A}}^{(t)} \right)^{-1} \boldsymbol{X}_t^\top \left( \boldsymbol{X}_t \boldsymbol{\theta}_\star + \boldsymbol{z}_t \right).
$$

Therefore, we can decompose $\left\| \hat{\boldsymbol{\theta}}_t - \boldsymbol{\theta}_\star \right\|_{\hat{\boldsymbol{A}}^{(t)}}^2$ into two parts as follows

$$
\left\| \hat{\boldsymbol{\theta}}_t - \boldsymbol{\theta}_\star \right\|_{\hat{\boldsymbol{A}}^{(t)}}^2
$$
$$
= \left( \hat{\boldsymbol{\theta}}_t - \boldsymbol{\theta}_\star \right)^\top \hat{\boldsymbol{A}}^{(t)} \left( \hat{\boldsymbol{\theta}}_t - \boldsymbol{\theta}_\star \right)
$$
$$
= \left( \hat{\boldsymbol{\theta}}_t - \boldsymbol{\theta}_\star \right)^\top \hat{\boldsymbol{A}}^{(t)} \left( \left( \hat{\boldsymbol{A}}^{(t)} \right)^{-1} \boldsymbol{X}_t^\top \left( \boldsymbol{X}_t \boldsymbol{\theta}_\star + \boldsymbol{z}_t \right) - \boldsymbol{\theta}_\star \right) \tag{11}
$$
$$
= \underbrace{\left( \hat{\boldsymbol{\theta}}_t - \boldsymbol{\theta}_\star \right)^\top \hat{\boldsymbol{A}}^{(t)} \left( \left( \hat{\boldsymbol{A}}^{(t)} \right)^{-1} \boldsymbol{X}_t^\top \boldsymbol{X}_t \boldsymbol{\theta}_\star - \boldsymbol{\theta}_\star \right)}_{\text{Term 1: Bias Error}} + \underbrace{\left( \hat{\boldsymbol{\theta}}_t - \boldsymbol{\theta}_\star \right)^\top \boldsymbol{X}_t^\top \boldsymbol{z}_t}_{\text{Term 2: Variance Error}}.
$$

**Bounding the bias error.**    We first focus on bounding the first term. We have that

$$
\left( \hat{\boldsymbol{\theta}}_t - \boldsymbol{\theta}_\star \right)^\top \hat{\boldsymbol{A}}^{(t)} \left( \left( \hat{\boldsymbol{A}}^{(t)} \right)^{-1} \boldsymbol{X}_t^\top \boldsymbol{X}_t \boldsymbol{\theta}_\star - \boldsymbol{\theta}_\star \right)
$$
$$
= \left( \hat{\boldsymbol{\theta}}_t - \boldsymbol{\theta}_\star \right)^\top \left( \hat{\boldsymbol{A}}^{(t)} \right)^{\frac{1}{2}} \left( \hat{\boldsymbol{A}}^{(t)} \right)^{-\frac{1}{2}} \left( \boldsymbol{X}_t^\top \boldsymbol{X}_t \boldsymbol{\theta}_\star - \hat{\boldsymbol{A}}^{(t)} \boldsymbol{\theta}_\star \right) \tag{12}
$$
$$
= \left( \hat{\boldsymbol{\theta}}_t - \boldsymbol{\theta}_\star \right)^\top \left( \hat{\boldsymbol{A}}^{(t)} \right)^{\frac{1}{2}} \left( \hat{\boldsymbol{A}}^{(t)} \right)^{-\frac{1}{2}} \left[ \left( \boldsymbol{A}^{(t)} - \hat{\boldsymbol{A}}^{(t)} \right) \boldsymbol{\theta}_\star - \lambda \boldsymbol{\theta}_\star \right].
$$

In accordance with the decomposability of matrix sketches, as detailed in Lemma 1, we have

$$\left\| \boldsymbol{X}_t^\top \boldsymbol{X}_t - \sum_{i=1}^{B_t} \left( \boldsymbol{S}_i^{(t)} \right)^\top \boldsymbol{S}_i^{(t)} \right\|_2 \leq \sum_{i=1}^{B_t} \overline{\sigma}_i \tag{13}$$

By Cauchy-Schwartz inequality and the triangle inequality, we have

$$\left( \hat{\boldsymbol{\theta}}_t - \boldsymbol{\theta}_\star \right)^\top \left( \hat{\boldsymbol{A}}^{(t)} \right)^{\frac{1}{2}} \left( \hat{\boldsymbol{A}}^{(t)} \right)^{-\frac{1}{2}} \left[ \left( \boldsymbol{A}^{(t)} - \hat{\boldsymbol{A}}^{(t)} \right) \boldsymbol{\theta}_\star - \lambda \boldsymbol{\theta}_\star \right]$$

$$\leq \left| \lambda + \sum_{i=1}^{B_t} \overline{\sigma}_i \right| \cdot \left\| \hat{\boldsymbol{\theta}}_t - \boldsymbol{\theta}_\star \right\|_{\hat{\boldsymbol{A}}^{(t)}} \cdot \| \boldsymbol{\theta}_\star \|_{\left( \hat{\boldsymbol{A}}^{(t)} \right)^{-1}} \tag{14}$$

$$\leq H \cdot \frac{\lambda + \sum_{i=1}^{B_t} \overline{\sigma}_i}{\sqrt{\lambda}} \cdot \left\| \hat{\boldsymbol{\theta}}_t - \boldsymbol{\theta}_\star \right\|_{\hat{\boldsymbol{A}}^{(t)}},$$

where the last inequality holds beacause $\hat{\boldsymbol{A}}^{(t)} \succeq \lambda \boldsymbol{I}$ and $\| \boldsymbol{\theta}_\star \|_2 \leq H$.

**Bounding the variance error.** Then, we aim to bound the second term. We use the following self-normalized martingale concentration inequality by Abbasi-Yadkori et al. (2011).

**Proposition 1.** *Assume that $\boldsymbol{z}_1, ..., \boldsymbol{z}_t$ is a conditionally $R$-subgaussian real-valued stochastic process and $\boldsymbol{X}_t^\top = \left[ \boldsymbol{x}_1^\top, ..., \boldsymbol{x}_t^\top \right]$ is any stochastic process such that $\boldsymbol{x}_i$ is measurable with respect to the $\sigma$-algebra generated by $\boldsymbol{z}_1, ..., \boldsymbol{z}_t$. Then, for any $\delta > 0$, with probability at least $1 - \delta$, for all $t \geq 0$,*

$$\left\| \boldsymbol{X}_t^\top \boldsymbol{z}_t \right\|_{\left( \boldsymbol{A}^{(t)} \right)^{-1}}^2 \leq 2R^2 \ln \left( \frac{1}{\delta} \left| \boldsymbol{A}^{(t)} \right|^{\frac{1}{2}} |\lambda \boldsymbol{I}|^{-\frac{1}{2}} \right).$$

Notice that the variance error can be reformulated as

$$\left( \hat{\boldsymbol{\theta}}_t - \boldsymbol{\theta}_\star \right)^\top \boldsymbol{X}_t^\top \boldsymbol{z}_t = \left( \hat{\boldsymbol{\theta}}_t - \boldsymbol{\theta}_\star \right)^\top \left( \boldsymbol{A}^{(t)} \right)^{-\frac{1}{2}} \left( \boldsymbol{A}^{(t)} \right)^{\frac{1}{2}} \boldsymbol{X}_t^\top \boldsymbol{z}_t$$

$$\leq \left\| \hat{\boldsymbol{\theta}}_t - \boldsymbol{\theta}_\star \right\|_{\hat{\boldsymbol{A}}^{(t)}} \cdot \frac{\left\| \hat{\boldsymbol{\theta}}_t - \boldsymbol{\theta}_\star \right\|_{\boldsymbol{A}^{(t)}}}{\left\| \hat{\boldsymbol{\theta}}_t - \boldsymbol{\theta}_\star \right\|_{\hat{\boldsymbol{A}}^{(t)}}} \cdot \left\| \boldsymbol{X}_t^\top \boldsymbol{z}_t \right\|_{\left( \boldsymbol{A}^{(t)} \right)^{-1}}, \tag{15}$$

where the last inequality uses Cauchy-Schwartz inequality.

For any vector $\boldsymbol{a}$, we have

$$\| \boldsymbol{a} \|_{\boldsymbol{A}^{(t)}}^2 - \| \boldsymbol{a} \|_{\hat{\boldsymbol{A}}^{(t)}}^2 = \boldsymbol{a}^\top \left( \boldsymbol{A}^{(t)} - \hat{\boldsymbol{A}}^{(t)} \right) \boldsymbol{a}$$

$$= \boldsymbol{a}^\top \left( \boldsymbol{X}^\top \boldsymbol{X} - \sum_{i=1}^{B_t} \left( \boldsymbol{S}_i^{(t)} \right)^\top \boldsymbol{S}_i^{(t)} \right) \boldsymbol{a} \tag{16}$$

$$\leq \sum_{i=1}^{B_t} \overline{\sigma}_i \cdot \| \boldsymbol{a} \|_2^2.$$

Therefore, the ratios of norms on the right-hand side of equation 15 can be bounded as

$$\frac{\left\| \hat{\boldsymbol{\theta}}_t - \boldsymbol{\theta}_\star \right\|_{\boldsymbol{A}^{(t)}}}{\left\| \hat{\boldsymbol{\theta}}_t - \boldsymbol{\theta}_\star \right\|_{\hat{\boldsymbol{A}}^{(t)}}} = \sqrt{\frac{\left\| \hat{\boldsymbol{\theta}}_t - \boldsymbol{\theta}_\star \right\|_{\boldsymbol{A}^{(t)}}^2}{\left\| \hat{\boldsymbol{\theta}}_t - \boldsymbol{\theta}_\star \right\|_{\hat{\boldsymbol{A}}^{(t)}}^2}}$$

$$\leq \sqrt{\frac{\left\| \hat{\boldsymbol{\theta}}_t - \boldsymbol{\theta}_\star \right\|_{\hat{\boldsymbol{A}}^{(t)}}^2 + \sum_{i=1}^{B_t} \overline{\sigma}_i \left\| \hat{\boldsymbol{\theta}}_t - \boldsymbol{\theta}_\star \right\|^2}{\left\| \hat{\boldsymbol{\theta}}_t - \boldsymbol{\theta}_\star \right\|_{\hat{\boldsymbol{A}}^{(t)}}^2}} \tag{17}$$

$$\leq \sqrt{1 + \frac{\sum_{i=1}^{B_t} \overline{\sigma}_i}{\lambda}}.$$

Substituting equation 17 and Proposition 1 into equation 15 gives

$$
\left\| \hat{\boldsymbol{\theta}}_t - \boldsymbol{\theta}_\star \right\|_{\hat{\boldsymbol{A}}^{(t)}} \cdot \frac{\left\| \hat{\boldsymbol{\theta}}_t - \boldsymbol{\theta}_\star \right\|_{\boldsymbol{A}^{(t)}}}{\left\| \hat{\boldsymbol{\theta}}_t - \boldsymbol{\theta}_\star \right\|_{\hat{\boldsymbol{A}}^{(t)}}} \cdot \left\| \boldsymbol{X}_t^\top \boldsymbol{z}_t \right\|_{\left(\boldsymbol{A}^{(t)}\right)^{-1}}
$$

$$
\leq \sqrt{1 + \frac{\sum_{i=1}^{B_t} \overline{\sigma}_i}{\lambda}} \cdot \sqrt{2R^2 \ln\left( \frac{1}{\delta} \left| \boldsymbol{A}^{(t)} \right|^{\frac{1}{2}} |\lambda \boldsymbol{I}|^{-\frac{1}{2}} \right)} \cdot \left\| \hat{\boldsymbol{\theta}}_t - \boldsymbol{\theta}_\star \right\|_{\hat{\boldsymbol{A}}^{(t)}} .
$$

(18)

Motivated by Abbasi-Yadkori et al. (2011); Kuzborskij et al. (2019), we apply the multi-scale sketch-based determinant-trace inequality. Compared to the non-sketched version, this inequality depends on the approximate covariance matrix $\hat{\boldsymbol{A}}$, reflecting the costs associated with the shrinkage due to multi-scale sketching.

**Lemma 2.** *For any $t \geq 1$, define $\boldsymbol{A}^{(t)} = \lambda \boldsymbol{I} + \boldsymbol{X}_t^\top \boldsymbol{X}_t$, and assume $\|\boldsymbol{x}_t\|_2 \leq L$, we have*

$$
\ln\left( \frac{\left| \boldsymbol{A}^{(t)} \right|}{|\lambda \boldsymbol{I}|} \right) \leq d \ln\left( 1 + \frac{\sum_{i=1}^{B_t} \overline{\sigma}_i}{\lambda} \right) + 2l_{B_t} \cdot \ln\left( 1 + \frac{tL^2}{2l_{B_t}\lambda} \right) .
$$

*Proof.* $\sum_{i=1}^{B_t} \left( \boldsymbol{S}_i^{(t)} \right)^\top \boldsymbol{S}_i^{(t)}$ has rank at most $2l_{B_t}$ due to the Dyadic Block Sketching. Since $\hat{\boldsymbol{A}}^{(t)} = \lambda \boldsymbol{I} + \sum_{i=1}^{B_t} \left( \boldsymbol{S}_i^{(t)} \right)^\top \boldsymbol{S}_i^{(t)}$ and $\boldsymbol{A}^{(t)} \preceq \hat{\boldsymbol{A}}^{(t)} + \sum_{i=1}^{B_t} \overline{\sigma}_i \cdot \boldsymbol{I}$, we have

$$
\left| \boldsymbol{A}^{(t)} \right| \leq \left| \hat{\boldsymbol{A}}^{(t)} + \sum_{i=1}^{B_t} \overline{\sigma}_i \cdot \boldsymbol{I} \right|
$$

$$
\leq \left( \lambda + \sum_{i=1}^{B_t} \overline{\sigma}_i \right)^{d - 2l_{B_t}} \cdot \left( \frac{\sum_{i=1}^{2l_{B_t}} \left( \lambda_i\left(\hat{\boldsymbol{A}}^{(t)}\right) + \sum_{i=1}^{B_t} \overline{\sigma}_i \right)}{2l_{B_t}} \right)^{2l_{B_t}}
$$

$$
\leq \left( \lambda + \sum_{i=1}^{B_t} \overline{\sigma}_i \right)^{d - 2l_{B_t}} \cdot \left( \lambda + \sum_{i=1}^{B_t} \overline{\sigma}_i + \frac{\mathrm{Tr}\left( \sum_{i=1}^{B_t} \left( \boldsymbol{S}_i^{(t)} \right)^\top \boldsymbol{S}_i^{(t)} \right)}{2l_{B_t}} \right)^{2l_{B_t}}
$$

$$
\leq \left( \lambda + \sum_{i=1}^{B_t} \overline{\sigma}_i \right)^{d - 2l_{B_t}} \cdot \left( \lambda + \sum_{i=1}^{B_t} \overline{\sigma}_i + \frac{tL^2}{2l_{B_t}} \right)^{2l_{B_t}} ,
$$

where the last inequality holds because

$$
\mathrm{Tr}\left( \sum_{i=1}^{B_t} \left( \boldsymbol{S}_i^{(t)} \right)^\top \boldsymbol{S}_i^{(t)} \right)
$$
$$
\leq \mathrm{Tr}\left( \boldsymbol{X}_t^\top \boldsymbol{X}_t \right)
$$
$$
= \sum_{s=1}^{t} \boldsymbol{x}_s^\top \boldsymbol{x}_s
$$
$$
\leq tL^2
$$

Therefore, we have

$$
\ln\left(\frac{\left|\boldsymbol{A}^{(t)}\right|}{|\lambda\boldsymbol{I}|}\right) \leq \ln\left\{\left(\frac{\lambda + \sum_{i=1}^{B_t}\overline{\sigma}_i}{\lambda}\right)^{d-2l_{B_t}} \cdot \left(\frac{\lambda + \sum_{i=1}^{B_t}\overline{\sigma}_i + \frac{tL^2}{2l_{B_t}}}{\lambda}\right)^{2l_{B_t}}\right\}
$$

$$
= (d - 2l_{B_t})\ln\left(1 + \frac{\sum_{i=1}^{B_t}\overline{\sigma}_i}{\lambda}\right) + 2l_{B_t}\ln\left(1 + \frac{\sum_{i=1}^{B_t}\overline{\sigma}_i}{\lambda} + \frac{tL^2}{2l_{B_t}\lambda}\right)
$$

$$
\leq d\ln\left(1 + \frac{\sum_{i=1}^{B_t}\overline{\sigma}_i}{\lambda}\right) + 2l_{B_t}\cdot\ln\left(1 + \frac{tL^2}{2l_{B_t}\lambda}\right).
$$

$\square$

According to Lemma 2, we finally bound the variance error term as follows

$$
\left\|\hat{\boldsymbol{\theta}}_t - \boldsymbol{\theta}_\star\right\|_{\hat{\boldsymbol{A}}^{(t)}} \cdot \frac{\left\|\hat{\boldsymbol{\theta}}_t - \boldsymbol{\theta}_\star\right\|_{\boldsymbol{A}^{(t)}}}{\left\|\hat{\boldsymbol{\theta}}_t - \boldsymbol{\theta}_\star\right\|_{\hat{\boldsymbol{A}}^{(t)}}} \cdot \left\|\boldsymbol{X}_t^\top\boldsymbol{z}_t\right\|_{(\boldsymbol{A}^{(t)})^{-1}}
$$

$$
\leq \sqrt{1 + \frac{\sum_{i=1}^{B_t}\overline{\sigma}_i}{\lambda}} \cdot \sqrt{2R^2\ln\left(\frac{1}{\delta}\left|\boldsymbol{A}^{(t)}\right|^{\frac{1}{2}}|\lambda\boldsymbol{I}|^{-\frac{1}{2}}\right)} \cdot \left\|\hat{\boldsymbol{\theta}}_t - \boldsymbol{\theta}_\star\right\|_{\hat{\boldsymbol{A}}^{(t)}}
$$

$$
\leq R \cdot \sqrt{1 + \frac{\sum_{i=1}^{B_t}\overline{\sigma}_i}{\lambda}} \cdot \sqrt{2\ln\left(\frac{1}{\delta}\right) + d\ln\left(1 + \frac{\sum_{i=1}^{B_t}\overline{\sigma}_i}{\lambda}\right) + 2l_{B_t}\cdot\ln\left(1 + \frac{tL^2}{2l_{B_t}\lambda}\right)} \cdot \left\|\hat{\boldsymbol{\theta}}_t - \boldsymbol{\theta}_\star\right\|_{\hat{\boldsymbol{A}}^{(t)}}.
$$

Sum up the bias error and the variance error and divide both sides of equation 11 by $\left\|\hat{\boldsymbol{\theta}}_t - \boldsymbol{\theta}_\star\right\|_{\hat{\boldsymbol{A}}^{(t)}}$ simultaneously, we have

$$
\left\|\hat{\boldsymbol{\theta}}_t - \boldsymbol{\theta}_\star\right\|_{\hat{\boldsymbol{A}}^{(t)}} \leq R \cdot \sqrt{1 + \frac{\sum_{i=1}^{B_t}\overline{\sigma}_i}{\lambda}} \cdot \sqrt{2\ln\left(\frac{1}{\delta}\right) + d\ln\left(1 + \frac{\sum_{i=1}^{B_t}\overline{\sigma}_i}{\lambda}\right) + 2l_{B_t}\cdot\ln\left(1 + \frac{tL^2}{2l_{B_t}\lambda}\right)}
$$

$$
+ H \cdot \frac{\lambda + \sum_{i=1}^{B_t}\overline{\sigma}_i}{\sqrt{\lambda}},
$$

which concludes the proof. $\square$

Having established the confidence ellipsoid, we now focus on analyzing the regret. We begin with an analysis of the instantaneous regret.

Recall that the optimal arm at round $t$ is defined as $\boldsymbol{x}_t^\star = \arg\max_{\boldsymbol{x}\in\mathcal{X}_t}(\boldsymbol{x}^\top\boldsymbol{\theta}_\star)$. On the other hand, the principle of optimism in the face of uncertainty ensures that $\left(\boldsymbol{x}_t, \hat{\boldsymbol{\theta}}_{t-1}\right) = \arg\max_{(\boldsymbol{x},\boldsymbol{\theta})\in\mathcal{X}_t\times\Theta_{t-1}} \boldsymbol{x}^\top\boldsymbol{\theta}$. By denoting $\tilde{\boldsymbol{\theta}}_t$ as the RLS estimator, we utilize these facts to establish the bound on the instantaneous regret as follows

$$
\begin{aligned}
&\left(\boldsymbol{x}_t^\star - \boldsymbol{x}_t\right)^\top\boldsymbol{\theta}_\star \\
&\leq \boldsymbol{x}_t^\top\hat{\boldsymbol{\theta}}_{t-1} - \boldsymbol{x}_t^\top\boldsymbol{\theta}_\star \\
&= \boldsymbol{x}_t^\top\left(\hat{\boldsymbol{\theta}}_{t-1} - \tilde{\boldsymbol{\theta}}_{t-1}\right) + \boldsymbol{x}_t^\top\left(\tilde{\boldsymbol{\theta}}_{t-1} - \boldsymbol{\theta}_\star\right) \\
&\leq \|\boldsymbol{x}_t\|_{(\hat{\boldsymbol{A}}^{(t-1)})^{-1}} \cdot \left(\left\|\hat{\boldsymbol{\theta}}_{t-1} - \tilde{\boldsymbol{\theta}}_{t-1}\right\|_{\hat{\boldsymbol{A}}^{(t-1)}} + \left\|\tilde{\boldsymbol{\theta}}_{t-1} - \boldsymbol{\theta}_\star\right\|_{\hat{\boldsymbol{A}}^{(t-1)}}\right) \\
&\leq 2\hat{\beta}_{t-1}(\delta) \cdot \|\boldsymbol{x}_t\|_{(\hat{\boldsymbol{A}}^{(t-1)})^{-1}}.
\end{aligned}
\tag{19}
$$

Now, we are prepared to establish the upper bound of regret. Utilizing equation 19 and Cauchy-Schwartz inequality, we derive the following bound

$$
\begin{aligned}
\text{Regret}_T &= \sum_{t=1}^{T} \max_{\boldsymbol{x} \in \mathcal{X}} \boldsymbol{x}^\top \boldsymbol{\theta}_\star - \sum_{t=1}^{T} \boldsymbol{x}_t^\top \boldsymbol{\theta}_\star \\
&\leq 2 \sum_{t=1}^{T} \min \left\{ HL, \hat{\beta}_{t-1}(\delta) \cdot \|\boldsymbol{x}_t\|_{(\hat{\boldsymbol{A}}^{(t-1)})^{-1}} \right\} \\
&\leq 2 \sum_{t=1}^{T} \hat{\beta}_{t-1}(\delta) \min \left\{ \frac{L}{\sqrt{\lambda}}, \|\boldsymbol{x}_t\|_{(\hat{\boldsymbol{A}}^{(t-1)})^{-1}} \right\} \\
&\leq 2 \cdot \max \left\{ 1, \frac{L}{\sqrt{\lambda}} \right\} \cdot \hat{\beta}_T(\delta) \cdot \sum_{t=1}^{T} \min \left\{ 1, \|\boldsymbol{x}_t\|_{(\hat{\boldsymbol{A}}^{(t-1)})^{-1}} \right\} \\
&\leq 2 \cdot \max \left\{ 1, \frac{L}{\sqrt{\lambda}} \right\} \cdot \hat{\beta}_T(\delta) \cdot \sqrt{T \sum_{t=1}^{T} \min \left\{ 1, \|\boldsymbol{x}_t\|_{(\hat{\boldsymbol{A}}^{(t-1)})^{-1}}^2 \right\}}.
\end{aligned}
\tag{20}
$$

We further bound the terms in the above. In particular, we formulate $\hat{\beta}_T(\delta)$ by Theorem 5 as follows

$$
\begin{aligned}
\hat{\beta}_T(\delta) &= R \sqrt{1 + \frac{\sum_{i=1}^{B_T} \overline{\sigma}_i}{\lambda}} \cdot \sqrt{2 \ln \frac{1}{\delta} + d \ln \left( 1 + \frac{\sum_{i=1}^{B_T} \overline{\sigma}_i}{\lambda} \right) + 2 l_{B_T} \cdot \ln \left( 1 + \frac{TL^2}{2 l_{B_T} \lambda} \right)} \\
&\quad + H\sqrt{\lambda} \left( 1 + \frac{\sum_{i=1}^{B_T} \overline{\sigma}_i}{\lambda} \right).
\end{aligned}
\tag{21}
$$

Besides, we adopt the Sketched leverage scores established by Kuzborskij et al. (2019) as follows

**Proposition 2** (Lemma 6 of Kuzborskij et al. (2019))**.** *The sketched leverage scores through sketching at round $T$ can be upper bounded as*

$$
\begin{aligned}
&\sum_{t=1}^{T} \min \left\{ 1, \|\boldsymbol{x}_t\|_{(\hat{\boldsymbol{A}}^{(t)})^{-1}}^2 \right\} \\
&\leq 2 \left( 1 + \frac{\sum_{i=1}^{B_T} \overline{\sigma}_i}{\lambda} \right) \cdot \ln \left( \frac{|\boldsymbol{A}^{(T)}|}{|\lambda \boldsymbol{I}|} \right) \\
&\leq 2 \left( 1 + \frac{\sum_{i=1}^{B_T} \overline{\sigma}_i}{\lambda} \right) \cdot \left( d \ln \left( \frac{1 + \sum_{i=1}^{B_T} \overline{\sigma}_i}{\lambda} \right) + 2 l_{B_t} \cdot \ln \left( 1 + \frac{TL^2}{2 l_{B_T} \lambda} \right) \right).
\end{aligned}
$$

Combining equation 21, equation 20 and Proposition 2, assume $L \geq \sqrt{\lambda}$, we have

$$\text{Regret}_T \leq 2 \cdot \max\left\{1, \frac{L}{\sqrt{\lambda}}\right\} \cdot \hat{\beta}_T(\delta) \cdot \sqrt{T \sum_{t=1}^{T} \min\left\{1, \|\boldsymbol{x}_t\|_{(\hat{\boldsymbol{A}}^{(t-1)})^{-1}}^2\right\}}$$

$$\stackrel{\tilde{O}}{=} \frac{L}{\sqrt{\lambda}} \cdot \sqrt{T} \cdot \left(1 + \frac{\sum_{i=1}^{B_T} \overline{\sigma}_i}{\lambda}\right) \cdot \left(d\ln\left(\frac{1 + \sum_{i=1}^{B_T} \overline{\sigma}_i}{\lambda}\right) + 2l_{B_t} \cdot \ln\left(1 + \frac{TL^2}{2l_{B_T}\lambda}\right)\right)$$

$$\cdot \left(R\sqrt{1 + \frac{\sum_{i=1}^{B_T} \overline{\sigma}_i}{\lambda}} \cdot \sqrt{2\ln\frac{1}{\delta} + d\ln\left(1 + \frac{\sum_{i=1}^{B_T} \overline{\sigma}_i}{\lambda}\right) + 2l_{B_T} \cdot \ln\left(1 + \frac{TL^2}{2l_{B_T}\lambda}\right)}\right.$$

$$\left. + H\sqrt{\lambda}\left(1 + \frac{\sum_{i=1}^{B_T} \overline{\sigma}_i}{\lambda}\right)\right)$$

$$\stackrel{\tilde{O}}{=} \frac{L(R + H\sqrt{\lambda})}{\sqrt{\lambda}} \cdot \left(d\ln\left(1 + \frac{\sum_{i=1}^{B_T} \overline{\sigma}_i}{\lambda}\right) + 2l_{B_T} \cdot \ln\left(1 + \frac{TL^2}{2l_{B_T}\lambda}\right)\right)$$

$$\cdot \left(1 + \frac{\sum_{i=1}^{B_T} \overline{\sigma}_i}{\lambda}\right)^{\frac{3}{2}} \sqrt{T}.$$

According to Theorem 2, we can bound the spectral error by error $\epsilon$, which is

$$\text{Regret}_T \stackrel{\tilde{O}}{=} \frac{L(R + H\sqrt{\lambda})}{\sqrt{\lambda}} \cdot \left(d\ln\left(1 + \frac{\epsilon}{\lambda}\right) + 2l_{B_T} \cdot \ln\left(1 + \frac{TL^2}{2l_{B_T}\lambda}\right)\right) \cdot \left(1 + \frac{\epsilon}{\lambda}\right)^{\frac{3}{2}} \sqrt{T}.$$

# F    PROOF OF THEOREM 4

We denote $B_t$ as the number of blocks at round $t$, and $\overline{\sigma}_i$ as the cumulative shrinking singular values in the sketch of block $i$. Let $l_{B_t}$ be the sketch size in the active block at round $t$. Similarly, our analysis establishes an intermediate result regarding the confidence ellipsoid.

**Theorem 6.** *Let $\hat{\boldsymbol{\theta}}_t$ be the RLS estimate constructed by an arbitrary policy for linear bandits after $t$ rounds of play. For any $\delta \in (0, 1)$, the optimal unknown weight $\boldsymbol{\theta}_\star$ belongs to the set $\Theta_t \equiv \left\{\boldsymbol{\theta} \in \mathbb{R}^d : \left\|\boldsymbol{\theta} - \hat{\boldsymbol{\theta}}_t\right\|_{\hat{\boldsymbol{A}}^{(t)}} \leq \hat{\beta}_t(\delta)\right\}$ with probability at least $1 - \delta$, where*

$$\hat{\beta}_t(\delta) = R \cdot \sqrt{2\ln\left(\frac{1}{\delta}\right) + d\ln\left(1 + \frac{\sum_{i=1}^{B_t} \overline{\sigma}_i}{\lambda}\right) + 2l_{B_t} \cdot \ln\left(1 + \frac{tL^2}{2l_{B_t}\lambda} + \frac{h_t}{\lambda}\right)}$$

$$+ H \cdot \sqrt{\lambda + \sum_{i=1}^{B_t} \overline{\sigma}_i}$$

*and*

$$h_t = \sum_{i=1}^{B_t} \overline{\sigma}_i - \frac{\sum_{i=1}^{B_t} l_i \cdot \overline{\sigma}_i}{2l_{B_t}}.$$

*Proof.* Notice that RFD uses the adaptive regularization term to approximate the covariance matrix, i.e., $\hat{\boldsymbol{A}}^{(t)} = \lambda\boldsymbol{I} + \sum_{i=1}^{B_t} \alpha_i^{(t)}\boldsymbol{I} + \sum_{i=1}^{B_t} \left(\boldsymbol{S}_i^{(t)}\right)^\top \boldsymbol{S}_i^{(t)}$, where $\boldsymbol{S}_i^{(t)}$ is the sketch matrix in block $i$ and $\alpha_i^{(t)}$ is the adaptive regularization term of RFD at round $t$.

Similarily, we decompose $\left\|\hat{\boldsymbol{\theta}}_t - \boldsymbol{\theta}_\star\right\|^2_{\hat{\boldsymbol{A}}^{(t)}}$ into two parts as follows

$$\left\|\hat{\boldsymbol{\theta}}_t - \boldsymbol{\theta}_\star\right\|^2_{\hat{\boldsymbol{A}}^{(t)}}$$

$$= \left(\hat{\boldsymbol{\theta}}_t - \boldsymbol{\theta}_\star\right)^\top \hat{\boldsymbol{A}}^{(t)} \left(\hat{\boldsymbol{\theta}}_t - \boldsymbol{\theta}_\star\right)$$

$$= \left(\hat{\boldsymbol{\theta}}_t - \boldsymbol{\theta}_\star\right)^\top \hat{\boldsymbol{A}}^{(t)} \left(\left(\hat{\boldsymbol{A}}^{(t)}\right)^{-1} \boldsymbol{X}_t^\top \left(\boldsymbol{X}_t\boldsymbol{\theta}_\star + \boldsymbol{z}_t\right) - \boldsymbol{\theta}_\star\right)$$

$$= \underbrace{\left(\hat{\boldsymbol{\theta}}_t - \boldsymbol{\theta}_\star\right)^\top \hat{\boldsymbol{A}}^{(t)} \left(\left(\hat{\boldsymbol{A}}^{(t)}\right)^{-1} \boldsymbol{X}_t^\top \boldsymbol{X}_t\boldsymbol{\theta}_\star - \boldsymbol{\theta}_\star\right)}_{\text{Term 1: Bias Error}} + \underbrace{\left(\hat{\boldsymbol{\theta}}_t - \boldsymbol{\theta}_\star\right)^\top \boldsymbol{X}_t^\top \boldsymbol{z}_t}_{\text{Term 2: Variance Error}}.$$

**Bounding the bias error.** For the bias error term, we have

$$\left(\hat{\boldsymbol{\theta}}_t - \boldsymbol{\theta}_\star\right)^\top \hat{\boldsymbol{A}}^{(t)} \left(\left(\hat{\boldsymbol{A}}^{(t)}\right)^{-1} \boldsymbol{X}_t^\top \boldsymbol{X}_t\boldsymbol{\theta}_\star - \boldsymbol{\theta}_\star\right)$$

$$= \left(\hat{\boldsymbol{\theta}}_t - \boldsymbol{\theta}_\star\right)^\top \left(\hat{\boldsymbol{A}}^{(t)}\right)^{\frac{1}{2}} \left(\hat{\boldsymbol{A}}^{(t)}\right)^{-\frac{1}{2}} \left(\boldsymbol{X}_t^\top \boldsymbol{X}_t\boldsymbol{\theta}_\star - \hat{\boldsymbol{A}}^{(t)}\boldsymbol{\theta}_\star\right)$$

$$= \left(\hat{\boldsymbol{\theta}}_t - \boldsymbol{\theta}_\star\right)^\top \left(\hat{\boldsymbol{A}}^{(t)}\right)^{\frac{1}{2}} \left(\hat{\boldsymbol{A}}^{(t)}\right)^{-\frac{1}{2}} \left(\boldsymbol{X}_t^\top \boldsymbol{X}_t - \lambda\boldsymbol{I} - \sum_{i=1}^{B_t} \alpha_i^{(t)}\boldsymbol{I} - \sum_{i=1}^{B_t} \left(\boldsymbol{S}_i^{(t)}\right)^\top \boldsymbol{S}_i^{(t)}\right)\boldsymbol{\theta}_\star \tag{22}$$

$$\triangleq \left(\hat{\boldsymbol{\theta}}_t - \boldsymbol{\theta}_\star\right)^\top \left(\hat{\boldsymbol{A}}^{(t)}\right)^{\frac{1}{2}} \left(\hat{\boldsymbol{A}}^{(t)}\right)^{-\frac{1}{2}} \boldsymbol{D}_t \cdot \boldsymbol{\theta}_\star$$

Since $\boldsymbol{D}_t = \boldsymbol{X}_t^\top \boldsymbol{X}_t - \lambda\boldsymbol{I} - \sum_{i=1}^{B_t} \alpha_i^{(t)}\boldsymbol{I} - \sum_{i=1}^{B_t} \left(\boldsymbol{S}_i^{(t)}\right)^\top \boldsymbol{S}_i^{(t)}$, for any unit vector $\boldsymbol{a}$, we have

$$\left|\boldsymbol{a}^\top \boldsymbol{D}_t\boldsymbol{a}\right| = \left|\boldsymbol{a}^\top \left(\boldsymbol{X}_t^\top \boldsymbol{X}_t - \lambda\boldsymbol{I} - \sum_{i=1}^{B_t} \alpha_i^{(t)}\boldsymbol{I} - \sum_{i=1}^{B_t} \left(\boldsymbol{S}_i^{(t)}\right)^\top \boldsymbol{S}_i^{(t)}\right)\boldsymbol{a}\right|$$

$$= \left|\boldsymbol{a}^\top \left(\boldsymbol{X}_t^\top \boldsymbol{X}_t - \sum_{i=1}^{B_t} \left(\boldsymbol{S}_i^{(t)}\right)^\top \boldsymbol{S}_i^{(t)}\right)\boldsymbol{a} - \lambda\boldsymbol{I} - \sum_{i=1}^{B_t} \alpha_i^{(t)}\boldsymbol{I}\right|. \tag{23}$$

According to Theroem 2, we can get

$$0 \le \boldsymbol{a}^\top \left(\boldsymbol{X}_t^\top \boldsymbol{X}_t - \sum_{i=1}^{B_t} \left(\boldsymbol{S}_i^{(t)}\right)^\top \boldsymbol{S}_i^{(t)}\right)\boldsymbol{a} \le \sum_{i=1}^{B_t} \overline{\sigma}_i.$$

Bring the above equation into equation 23, since $\sum_{i=1}^{B_t} \alpha_i^{(t)} = \sum_{i=1}^{B_t} \overline{\sigma}_i$, we can bound the spectral norm of $\boldsymbol{D}_t$ as follows

$$\|\boldsymbol{D}_t\|_2 \le \lambda + \sum_{i=1}^{B_t} \overline{\sigma}_i. \tag{24}$$

By Cauchy-Schwartz inequality and the triangle inequality, we can bound equation 22 by

$$\left(\hat{\boldsymbol{\theta}}_t - \boldsymbol{\theta}_\star\right)^\top \left(\hat{\boldsymbol{A}}^{(t)}\right)^{\frac{1}{2}} \left(\hat{\boldsymbol{A}}^{(t)}\right)^{-\frac{1}{2}} \boldsymbol{D}_t \cdot \boldsymbol{\theta}_\star$$

$$\le \left\|\hat{\boldsymbol{\theta}}_t - \boldsymbol{\theta}_\star\right\|_{\hat{\boldsymbol{A}}^{(t)}} \cdot \|\boldsymbol{D}_t\|_2 \cdot \|\boldsymbol{\theta}_\star\|_{\left(\hat{\boldsymbol{A}}^{(t)}\right)^{-1}} \tag{25}$$

$$\le H \cdot \sqrt{\lambda + \sum_{i=1}^{B_t} \overline{\sigma}_i} \cdot \left\|\hat{\boldsymbol{\theta}}_t - \boldsymbol{\theta}_\star\right\|_{\hat{\boldsymbol{A}}^{(t)}},$$

where the last inequality holds because

$$\|\boldsymbol{\theta}_\star\|^2_{\left(\hat{\boldsymbol{A}}^{(t)}\right)^{-1}} \le \frac{\|\boldsymbol{\theta}_\star\|^2_2}{\lambda_{\min}\left(\hat{\boldsymbol{A}}^{(t)}\right)} \le \frac{H^2}{\lambda + \sum_{i=1}^{B_t} \overline{\sigma}_i}.$$

**Bounding the variance error.** For the variance error, we have

$$
\left(\hat{\boldsymbol{\theta}}_t - \boldsymbol{\theta}_\star\right)^\top \boldsymbol{X}_t^\top \boldsymbol{z}_t = \left(\hat{\boldsymbol{\theta}}_t - \boldsymbol{\theta}_\star\right)^\top \left(\boldsymbol{A}^{(t)}\right)^{-\frac{1}{2}} \left(\boldsymbol{A}^{(t)}\right)^{\frac{1}{2}} \boldsymbol{X}_t^\top \boldsymbol{z}_t
$$

$$
\leq \left\|\hat{\boldsymbol{\theta}}_t - \boldsymbol{\theta}_\star\right\|_{\hat{\boldsymbol{A}}^{(t)}} \cdot \frac{\left\|\hat{\boldsymbol{\theta}}_t - \boldsymbol{\theta}_\star\right\|_{\boldsymbol{A}^{(t)}}}{\left\|\hat{\boldsymbol{\theta}}_t - \boldsymbol{\theta}_\star\right\|_{\hat{\boldsymbol{A}}^{(t)}}} \cdot \left\|\boldsymbol{X}_t^\top \boldsymbol{z}_t\right\|_{\left(\boldsymbol{A}^{(t)}\right)^{-1}} \tag{26}
$$

$$
\leq \left\|\hat{\boldsymbol{\theta}}_t - \boldsymbol{\theta}_\star\right\|_{\hat{\boldsymbol{A}}^{(t)}} \cdot \left\|\boldsymbol{X}_t^\top \boldsymbol{z}_t\right\|_{\left(\boldsymbol{A}^{(t)}\right)^{-1}}.
$$

where the last inequality holds because for any vector $\boldsymbol{a}$

$$
\|\boldsymbol{a}\|_{\boldsymbol{A}^{(t)}}^2 - \|\boldsymbol{a}\|_{\hat{\boldsymbol{A}}^{(t)}}^2 = \boldsymbol{a}^\top \left(\boldsymbol{X}^\top \boldsymbol{X} - \sum_{i=1}^{B_t} \left(\boldsymbol{S}_i^{(t)}\right)^\top \boldsymbol{S}_i^{(t)} - \sum_{i=1}^{B_t} \overline{\sigma}_i \boldsymbol{I}\right) \boldsymbol{a}
$$

$$
= \boldsymbol{a}^\top \left(\boldsymbol{X}^\top \boldsymbol{X} - \sum_{i=1}^{B_t} \left(\boldsymbol{S}_i^{(t)}\right)^\top \boldsymbol{S}_i^{(t)}\right) \boldsymbol{a} - \sum_{i=1}^{B_t} \overline{\sigma}_i \|\boldsymbol{a}\|_2^2 \tag{27}
$$

$$
\leq \sum_{i=1}^{B_t} \overline{\sigma}_i \|\boldsymbol{a}\|_2^2 - \sum_{i=1}^{B_t} \overline{\sigma}_i \|\boldsymbol{a}\|_2^2
$$

$$
= 0
$$

By Proposition 1, we can bound the variance error term as follows

$$
\left(\hat{\boldsymbol{\theta}}_t - \boldsymbol{\theta}_\star\right)^\top \boldsymbol{X}_t^\top \boldsymbol{z}_t
$$

$$
= \left(\hat{\boldsymbol{\theta}}_t - \boldsymbol{\theta}_\star\right)^\top \left(\boldsymbol{A}^{(t)}\right)^{-\frac{1}{2}} \left(\boldsymbol{A}^{(t)}\right)^{\frac{1}{2}} \boldsymbol{X}_t^\top \boldsymbol{z}_t
$$

$$
\leq \left\|\hat{\boldsymbol{\theta}}_t - \boldsymbol{\theta}_\star\right\|_{\hat{\boldsymbol{A}}^{(t)}} \cdot \left\|\boldsymbol{X}_t^\top \boldsymbol{z}_t\right\|_{\left(\boldsymbol{A}^{(t)}\right)^{-1}} \tag{28}
$$

$$
\leq \sqrt{2R^2 \ln\left(\frac{1}{\delta} \left|\boldsymbol{A}^{(t)}\right|^{\frac{1}{2}} \left|\lambda \boldsymbol{I}\right|^{-\frac{1}{2}}\right)} \cdot \left\|\hat{\boldsymbol{\theta}}_t - \boldsymbol{\theta}_\star\right\|_{\hat{\boldsymbol{A}}^{(t)}}.
$$

According to equation 27, we have $\left|\hat{\boldsymbol{A}}^{(t)}\right| \geq \left|\boldsymbol{A}^{(t)}\right|$. For any $t \in [T]$, since the rank of $\hat{\boldsymbol{A}}^{(t)}$ is at most $2l_{B_t}$, we can bound the determinant of $\hat{\boldsymbol{A}}^{(t)}$ as follows

$$
\left|\hat{\boldsymbol{A}}^{(t)}\right| \leq \left(\sum_{i=1}^{B_t} \alpha_i^{(t)} + \lambda\right)^{d - 2l_{B_t}} \cdot \prod_{i=1}^{2l_{B_t}} \lambda_i\left(\hat{\boldsymbol{A}}^{(t)}\right)
$$

$$
\leq \left(\sum_{i=1}^{B_t} \alpha_i^{(t)} + \lambda\right)^{d - 2l_{B_t}} \left(\frac{\sum_{i=1}^{2l_{B_t}} \lambda_i\left(\hat{\boldsymbol{A}}^{(t)}\right)}{2l_{B_t}}\right)^{2l_{B_t}}
$$

$$
= \left(\sum_{i=1}^{B_t} \overline{\sigma}_i + \lambda\right)^{d - 2l_{B_t}} \left[\sum_{i=1}^{B_t} \overline{\sigma}_i + \lambda + \frac{\mathrm{Tr}\left(\sum_{i=1}^{B_t} \left(\boldsymbol{S}_i^{(t)}\right)^\top \boldsymbol{S}_i^{(t)}\right)}{2l_{B_t}}\right]^{2l_{B_t}} \tag{29}
$$

$$
\leq \left(\sum_{i=1}^{B_t} \overline{\sigma}_i + \lambda\right)^{d - 2l_{B_t}} \left(\left(\sum_{i=1}^{B_t} \overline{\sigma}_i - \frac{\sum_{i=1}^{B_t} l_i \cdot \overline{\sigma}_i}{2l_{B_t}}\right) + \lambda + \frac{TL^2}{2l_{B_t}}\right)^{2l_{B_t}},
$$

where the last inequality satisfies due to

$$\text{Tr}\left(\sum_{i=1}^{B_t}\left(\boldsymbol{S}_i^{(t)}\right)^\top \boldsymbol{S}_i^{(t)}\right) = \sum_{i=1}^{B_t}\text{Tr}\left(\left(\boldsymbol{S}_i^{(t)}\right)^\top \boldsymbol{S}_i^{(t)}\right)$$

$$= \sum_{s=1}^{t}\text{Tr}(\boldsymbol{x}_s^\top \boldsymbol{x}_s) - \sum_{i=1}^{B_t} l_i \cdot \overline{\sigma}_i$$

$$\leq TL^2 - \sum_{i=1}^{B_t} l_i \cdot \overline{\sigma}_i.$$

Therefore, the variance error term can be bounded as

$$\left(\hat{\boldsymbol{\theta}}_t - \boldsymbol{\theta}_\star\right)^\top \boldsymbol{X}_t^\top \boldsymbol{z}_t$$

$$\leq \sqrt{2R^2 \ln\left(\frac{1}{\delta}\left|\boldsymbol{A}^{(t)}\right|^{\frac{1}{2}}|\lambda\boldsymbol{I}|^{-\frac{1}{2}}\right)} \cdot \left\|\hat{\boldsymbol{\theta}}_t - \boldsymbol{\theta}_\star\right\|_{\hat{\boldsymbol{A}}^{(t)}}$$

$$\leq \sqrt{2R^2 \ln\left(\frac{1}{\delta}\left|\hat{\boldsymbol{A}}^{(t)}\right|^{\frac{1}{2}}|\lambda\boldsymbol{I}|^{-\frac{1}{2}}\right)} \cdot \left\|\hat{\boldsymbol{\theta}}_t - \boldsymbol{\theta}_\star\right\|_{\hat{\boldsymbol{A}}^{(t)}}$$

$$\leq R \cdot \sqrt{2\ln\left(\frac{1}{\delta}\right) + (d - 2l_{B_t})\ln\left(1 + \frac{\sum_{i=1}^{B_t}\overline{\sigma}_i}{\lambda}\right) + 2l_{B_t}\cdot\ln\left(1 + \frac{tL^2}{2l_{B_t}\lambda} + \frac{h_t}{\lambda}\right)} \cdot \left\|\hat{\boldsymbol{\theta}}_t - \boldsymbol{\theta}_\star\right\|_{\hat{\boldsymbol{A}}^{(t)}}$$

$$\leq R \cdot \sqrt{2\ln\left(\frac{1}{\delta}\right) + d\ln\left(1 + \frac{\sum_{i=1}^{B_t}\overline{\sigma}_i}{\lambda}\right) + 2l_{B_t}\cdot\ln\left(1 + \frac{tL^2}{2l_{B_t}\lambda} + \frac{h_t}{\lambda}\right)} \cdot \left\|\hat{\boldsymbol{\theta}}_t - \boldsymbol{\theta}_\star\right\|_{\hat{\boldsymbol{A}}^{(t)}},$$

where $h_t = \sum_{i=1}^{B_t}\overline{\sigma}_i - \frac{\sum_{i=1}^{B_t} l_i \cdot \overline{\sigma}_i}{2l_{B_t}}$.

Sum up the bias error term and the variance error term and divide both sides by $\left\|\hat{\boldsymbol{\theta}}_t - \boldsymbol{\theta}_\star\right\|_{\hat{\boldsymbol{A}}^{(t)}}$ simultaneously, we have

$$\left\|\hat{\boldsymbol{\theta}}_t - \boldsymbol{\theta}_\star\right\|_{\hat{\boldsymbol{A}}^{(t)}} \leq R \cdot \sqrt{2\ln\left(\frac{1}{\delta}\right) + d\ln\left(1 + \frac{\sum_{i=1}^{B_t}\overline{\sigma}_i}{\lambda}\right) + 2l_{B_t}\cdot\ln\left(1 + \frac{tL^2}{2l_{B_t}\lambda} + \frac{h_t}{\lambda}\right)}$$

$$+ H \cdot \sqrt{\lambda + \sum_{i=1}^{B_t}\overline{\sigma}_i},$$

which concludes the proof. $\qquad\square$

Next, we start to prove the regret. Similar to the case using FD, since the algorithm uses the principle of optimism in the face of uncertainty to select the arm, we can bound instantaneous regret by

equation 19. Utilizing equation 19 and Cauchy-Schwartz inequality, we derive the following bound

$$\text{Regret}_T = \sum_{t=1}^{T} \max_{\boldsymbol{x} \in \mathcal{X}} \boldsymbol{x}^\top \boldsymbol{\theta}_\star - \sum_{t=1}^{T} \boldsymbol{x}_t^\top \boldsymbol{\theta}_\star$$

$$\leq 2 \sum_{t=1}^{T} \min \left\{ HL, \hat{\beta}_{t-1}(\delta) \cdot \|\boldsymbol{x}_t\|_{\left(\hat{\boldsymbol{A}}^{(t-1)}\right)^{-1}} \right\}$$

$$\leq 2 \sum_{t=1}^{T} \hat{\beta}_{t-1}(\delta) \min \left\{ \frac{L}{\sqrt{\lambda}}, \|\boldsymbol{x}_t\|_{\left(\hat{\boldsymbol{A}}^{(t-1)}\right)^{-1}} \right\} \tag{30}$$

$$\leq 2 \cdot \max \left\{ 1, \frac{L}{\sqrt{\lambda}} \right\} \cdot \hat{\beta}_T(\delta) \cdot \sum_{t=1}^{T} \min \left\{ 1, \|\boldsymbol{x}_t\|_{\left(\hat{\boldsymbol{A}}^{(t-1)}\right)^{-1}} \right\}$$

$$\leq 2 \cdot \max \left\{ 1, \frac{L}{\sqrt{\lambda}} \right\} \cdot \hat{\beta}_T(\delta) \cdot \sqrt{T \sum_{t=1}^{T} \min \left\{ 1, \|\boldsymbol{x}_t\|^2_{\left(\hat{\boldsymbol{A}}^{(t-1)}\right)^{-1}} \right\}}.$$

We present a lemma of RFD-sketched leverage scores to conclude the proof.

**Lemma 3.**

$$\sum_{t=1}^{T} \min \left\{ 1, \|\boldsymbol{x}_t\|^2_{\left(\hat{\boldsymbol{A}}^{(t-1)}\right)^{-1}} \right\} \leq 2l_{B_T} \cdot \ln \left( 1 + \frac{TL^2}{2l_{B_T}\lambda} + \frac{h_T}{\lambda} \right).$$

*Proof.* Denote $\boldsymbol{C}_t = \hat{\boldsymbol{A}}^{(t-1)} + \boldsymbol{x}_t^\top \boldsymbol{x}_t$. Notice that the first $2l_{B_t}$ eigenvalues of $\boldsymbol{C}_t$ are the same as $\hat{\boldsymbol{A}}^{(t)}$ while the other eigenvalues of $\boldsymbol{C}_t$ are $\sum_{i=1}^{B_t} \alpha_i^{(t-1)} + \lambda$. Thus we can obtain $\frac{|\hat{\boldsymbol{A}}^{(t)}|}{|\boldsymbol{C}_t|} = \left( \frac{\sum_{i=1}^{B_t} \alpha_i^{(t)} + \lambda}{\sum_{i=1}^{B_{t-1}} \alpha_i^{(t-1)} + \lambda} \right)^{d - 2l_{B_t}}$.

For the determinant of $\hat{\boldsymbol{A}}^{(t)}$, we have

$$\left| \hat{\boldsymbol{A}}^{(t)} \right| = \left( \frac{\sum_{i=1}^{B_t} \alpha_i^{(t)} + \lambda}{\sum_{i=1}^{B_{t-1}} \alpha_i^{(t-1)} + \lambda} \right)^{d - 2l_{B_t}} \cdot |\boldsymbol{C}_t|$$

$$= \left( \frac{\sum_{i=1}^{B_t} \alpha_i^{(t)} + \lambda}{\sum_{i=1}^{B_{t-1}} \alpha_i^{(t-1)} + \lambda} \right)^{d - 2l_{B_t}} \cdot \left| \hat{\boldsymbol{A}}^{(t-1)} \right| \cdot \left| \boldsymbol{I} + \left( \hat{\boldsymbol{A}}^{(t-1)} \right)^{-1} \boldsymbol{x}_t^\top \boldsymbol{x}_t \right| \tag{31}$$

$$= \left( \frac{\sum_{i=1}^{B_t} \alpha_i^{(t)} + \lambda}{\sum_{i=1}^{B_{t-1}} \alpha_i^{(t-1)} + \lambda} \right)^{d - 2l_{B_t}} \cdot \left| \hat{\boldsymbol{A}}^{(t-1)} \right| \cdot \left( 1 + \|\boldsymbol{x}_t\|^2_{\left(\hat{\boldsymbol{A}}^{(t-1)}\right)^{-1}} \right)$$

$$= \left( \frac{\sum_{i=1}^{B_t} \overline{\sigma}_i + \lambda}{\lambda} \right)^{d - 2l_{B_t}} \cdot |\lambda \boldsymbol{I}| \cdot \prod_{s=1}^{t} \left( 1 + \|\boldsymbol{x}_s\|^2_{\left(\hat{\boldsymbol{A}}^{(s-1)}\right)^{-1}} \right).$$

Since $\min(1, x) \leq 2\ln(1 + x)$ for all $x \geq 0$, using equation 31, we can derive the following bound

$$\sum_{t=1}^{T} \min \left\{ 1, \|\boldsymbol{x}_t\|^2_{\left(\hat{\boldsymbol{A}}^{(t-1)}\right)^{-1}} \right\}$$

$$\leq 2 \sum_{t=1}^{T} \ln \left( 1 + \|\boldsymbol{x}_t\|^2_{\left(\hat{\boldsymbol{A}}^{(t-1)}\right)^{-1}} \right)$$

$$= 2 \cdot \ln \left( \left( \frac{\lambda}{\sum_{i=1}^{B_T} \overline{\sigma}_i + \lambda} \right)^{d - 2l_{B_T}} \cdot \frac{\left| \hat{\boldsymbol{A}}^{(T)} \right|}{|\lambda \boldsymbol{I}|} \right)$$

$$\leq 2l_{B_T} \cdot \ln \left( 1 + \frac{TL^2}{2l_{B_T}\lambda} + \frac{h_T}{\lambda} \right),$$

where the last step holds by equation 29 and $h_T = \sum_{i=1}^{B_T} \overline{\sigma}_i - \frac{\sum_{i=1}^{B_T} l_i \cdot \overline{\sigma}_i}{2 l_{B_T}}$. $\qquad\square$

We combine equation 30, Theorem 6 and Lemma 3. Assume $L \geq \sqrt{\lambda}$, we have

$$
\text{Regret}_T = \sum_{t=1}^{T} \max_{\boldsymbol{x} \in \mathcal{X}} \boldsymbol{x}^\top \boldsymbol{\theta}_\star - \sum_{t=1}^{T} \boldsymbol{x}_t^\top \boldsymbol{\theta}_\star
$$

$$
\leq 2 \cdot \max\left\{1, \frac{L}{\sqrt{\lambda}}\right\} \cdot \hat{\beta}_T(\delta) \cdot \sqrt{T \sum_{t=1}^{T} \min\left\{1, \|\boldsymbol{x}_t\|^2_{(\hat{\boldsymbol{A}}^{(t-1)})^{-1}}\right\}}
$$

$$
\overset{\tilde{\mathcal{O}}}{=} \frac{L}{\sqrt{\lambda}} \cdot \sqrt{T} \cdot \sqrt{2 l_{B_T} \cdot \ln\left(1 + \frac{T L^2}{2 l_{B_T} \lambda} + \frac{h_T}{\lambda}\right)} \cdot \left(H \cdot \sqrt{\lambda + \sum_{i=1}^{B_T} \overline{\sigma}_i} + \right.
$$

$$
\left. R \cdot \sqrt{d \ln\left(1 + \frac{\sum_{i=1}^{B_T} \overline{\sigma}_i}{\lambda}\right) + 2 l_{B_T} \cdot \ln\left(1 + \frac{T L^2}{2 l_{B_T} \lambda} + \frac{h_T}{\lambda}\right)}\right).
$$

According to Theorem 2, we can bound the spectral error by error $\epsilon$, which is

$$
\text{Regret}_T \overset{\tilde{\mathcal{O}}}{=} \frac{L}{\sqrt{\lambda}} \cdot \sqrt{\ln\left(1 + \frac{T L^2}{2 l_{B_T} \lambda} + \frac{h_T}{\lambda}\right)} \cdot \left(H \cdot \sqrt{\lambda + \epsilon} + \right.
$$

$$
\left. R \cdot \sqrt{d \ln\left(1 + \frac{\epsilon}{\lambda}\right) + 2 l_{B_T} \cdot \ln\left(1 + \frac{T L^2}{2 l_{B_T} \lambda} + \frac{h_T}{\lambda}\right)}\right)
$$

$$
\overset{\tilde{\mathcal{O}}}{=} \frac{L}{\sqrt{\lambda}} \cdot \sqrt{\ln\left(1 + \frac{T L^2}{2 l_{B_T} \lambda} + \left(1 - \frac{2 - B_T}{2^{B_T + 1}}\right) \cdot \frac{\epsilon}{\lambda}\right)} \cdot \left(H \cdot \sqrt{\lambda + \epsilon} + \right.
$$

$$
\left. R \cdot \sqrt{d \ln\left(1 + \frac{\epsilon}{\lambda}\right) + 2 l_{B_T} \cdot \ln\left(1 + \frac{T L^2}{2 l_{B_T} \lambda} + \left(1 - \frac{2 - B_T}{2^{B_T + 1}}\right) \cdot \frac{\epsilon}{\lambda}\right)}\right),
$$

where the last inequality holds because

$$
h_T = \sum_{i=1}^{B_T} \overline{\sigma}_i - \frac{\sum_{i=1}^{B_T} l_i \cdot \overline{\sigma}_i}{2 l_{B_T}}
$$

$$
= \sum_{i=1}^{B_T} \left(1 - \frac{2^{i-1}}{2^{B_T}}\right) \cdot \overline{\sigma}_i
$$

$$
\leq \epsilon \cdot \sum_{i=1}^{B_T} \left(1 - \frac{2^{i-1}}{2^{B_T}}\right) \cdot \frac{1}{2^i} = \left(1 - \frac{2 - B_T}{2^{B_T + 1}}\right) \cdot \epsilon
$$

## G  PROPERTIES OF DYADIC BLOCK SKETCHING FOR RFD

In this section, we highlight two significant properties of Dyadic Block Sketching for RFD that elucidate why the regret bound of DBSLinUCB using RFD is improved. Although Robust Frequent Directions for ridge regression have been studied by Luo et al. (2019), their theory is limited to single deterministic streaming sketches. We demonstrate that the decomposability of multi-scale sketching does not alter the properties of RFD.

We begin with the positive definite monotonicity of Dyadic Block Sketching for RFD, which ensures that the sequence of approximation matrices is per-step optimal.

**Theorem 7** (Positive Definite Monotonicity). *At round $t$, denote that the Dyadic Block Sketching for RFD provides a sketch $\boldsymbol{S}^{(t)}$, we have the following equation*

$$
\left(\boldsymbol{S}^{(t)}\right)^\top \boldsymbol{S}^{(t)} + \alpha^{(t)} \boldsymbol{I} \succeq \left(\boldsymbol{S}^{(t-1)}\right)^\top \boldsymbol{S}^{(t-1)} + \alpha^{(t-1)} \boldsymbol{I}.
$$

*Proof.* Notice that $\alpha^{(t)}\boldsymbol{I} + \left(\boldsymbol{S}^{(t)}\right)^{\top}\boldsymbol{S}^{(t)} = \sum_{i=1}^{B_t}\alpha_i^{(t)}\boldsymbol{I} + \sum_{i=1}^{B_t}\left(\boldsymbol{S}_i^{(t)}\right)^{\top}\boldsymbol{S}_i^{(t)}$, where $\boldsymbol{S}_i^{(t)}$ is the sketch matrix in block $i$ and $\alpha_i^{(t)}$ is the adaptive regularization term of RFD at round $t$.

Let $\boldsymbol{Q} = \left[\left(\boldsymbol{S}_{B_t}^{(t-1)}\right)^{\top}, \boldsymbol{x}_t^{\top}\right]^{\top}$, $\sigma_t$ is the shrinking singular values of active block at round $t$, the shrinking step of RFD provides

$$\sum_{i=1}^{B_t}\left(\boldsymbol{S}_i^{(t)}\right)^{\top}\boldsymbol{S}_i^{(t)} + \sigma_t\boldsymbol{I} \succeq \sum_{i=1}^{B_t-1}\left(\boldsymbol{S}_i^{(t)}\right)^{\top}\boldsymbol{S}_i^{(t)} + \boldsymbol{Q}^{\top}\boldsymbol{Q} \succeq \sum_{i=1}^{B_t-1}\left(\boldsymbol{S}_i^{(t-1)}\right)^{\top}\boldsymbol{S}_i^{(t-1)}. \tag{32}$$

Therefore, for any unit vector $\boldsymbol{a}$, we have

$$\boldsymbol{a}^{\top}\left(\left(\boldsymbol{S}^{(t)}\right)^{\top}\boldsymbol{S}^{(t)} + \alpha^{(t)}\boldsymbol{I} - \left(\boldsymbol{S}^{(t-1)}\right)^{\top}\boldsymbol{S}^{(t-1)} + \alpha^{(t-1)}\boldsymbol{I}\right)\boldsymbol{a}$$

$$= \boldsymbol{a}^{\top}\left(\sum_{i=1}^{B_t}\alpha_i^{(t)}\boldsymbol{I} + \sum_{i=1}^{B_t}\left(\boldsymbol{S}_i^{(t)}\right)^{\top}\boldsymbol{S}_i^{(t)} - \sum_{i=1}^{B_{t-1}}\alpha_i^{(t-1)}\boldsymbol{I} - \sum_{i=1}^{B_{t-1}}\left(\boldsymbol{S}_i^{(t-1)}\right)^{\top}\boldsymbol{S}_i^{(t-1)}\right)\boldsymbol{a}$$

$$= \boldsymbol{a}^{\top}\left(\sum_{i=1}^{B_t}\left(\boldsymbol{S}_i^{(t)}\right)^{\top}\boldsymbol{S}_i^{(t)} + \sigma_t\boldsymbol{I} - \sum_{i=1}^{B_{t-1}}\left(\boldsymbol{S}_i^{(t-1)}\right)^{\top}\boldsymbol{S}_i^{(t-1)}\right)\boldsymbol{a}$$

$$\geq 0,$$

which concludes the proof. $\qquad\square$

Next, we prove that the sketch matrix produced by Dyadic Block Sketching for RFD is better conditioned than those produced by Dyadic Block Sketching for FD and the covariance matrix. In this context, the $\alpha$ selected by RFD is optimal, as choosing a smaller $\alpha$ would result in a worse condition number for the approximation matrices.

**Theorem 8** (Well-Conditioned Property). *Let* $\mathrm{cond}(\boldsymbol{X}) = \frac{\sigma_{\max}(\boldsymbol{X})}{\sigma_{\min}(\boldsymbol{X})}$ *be the condition number of matrix* $\boldsymbol{X}$. *At round* $t$, *denote that the Dyadic Block Sketching for RFD provides a sketch* $\boldsymbol{S}^{(t)}$, *we have*

$$\mathrm{cond}\left(\left(\boldsymbol{S}^{(t)}\right)^{\top}\boldsymbol{S}^{(t)} + \alpha^{(t)}\boldsymbol{I} + \lambda\boldsymbol{I}\right) \leq \mathrm{cond}\left(\left(\boldsymbol{S}^{(t)}\right)^{\top}\boldsymbol{S}^{(t)} + \lambda\boldsymbol{I}\right),$$

$$\mathrm{cond}\left(\left(\boldsymbol{S}^{(t)}\right)^{\top}\boldsymbol{S}^{(t)} + \alpha^{(t)}\boldsymbol{I} + \lambda\boldsymbol{I}\right) \leq \mathrm{cond}\left(\boldsymbol{X}_t^{\top}\boldsymbol{X}_t + \lambda\boldsymbol{I}\right).$$

*Proof.* Notice that $\alpha^{(t)}\boldsymbol{I} + \left(\boldsymbol{S}^{(t)}\right)^{\top}\boldsymbol{S}^{(t)} = \sum_{i=1}^{B_t}\alpha_i^{(t)}\boldsymbol{I} + \sum_{i=1}^{B_t}\left(\boldsymbol{S}_i^{(t)}\right)^{\top}\boldsymbol{S}_i^{(t)}$, where $\boldsymbol{S}_i^{(t)}$ is the sketch matrix in block $i$ and $\alpha_i^{(t)}$ is the adaptive regularization term of RFD at round $t$. We have

$$\mathrm{cond}\left(\left(\boldsymbol{S}^{(t)}\right)^{\top}\boldsymbol{S}^{(t)} + \alpha^{(t)}\boldsymbol{I} + \lambda\boldsymbol{I}\right) = \frac{\sigma_{\max}\left(\sum_{i=1}^{B_t}\left(\boldsymbol{S}_i^{(t)}\right)^{\top}\boldsymbol{S}_i^{(t)}\right) + \lambda + \sum_{i=1}^{B_t}\alpha_i^{(t)}}{\lambda + \sum_{i=1}^{B_t}\alpha_i^{(t)}}$$

$$\leq \frac{\sigma_{\max}\left(\sum_{i=1}^{B_t}\left(\boldsymbol{S}_i^{(t)}\right)^{\top}\boldsymbol{S}_i^{(t)}\right) + \lambda}{\lambda}$$

$$= \mathrm{cond}\left(\left(\boldsymbol{S}^{(t)}\right)^{\top}\boldsymbol{S}^{(t)} + \lambda\boldsymbol{I}\right).$$

Similarly, we have

$$\text{cond}\left(\left(\boldsymbol{S}^{(t)}\right)^{\top}\boldsymbol{S}^{(t)} + \alpha^{(t)}\boldsymbol{I} + \lambda\boldsymbol{I}\right) = \frac{\sigma_{\max}\left(\sum_{i=1}^{B_t}\left(\boldsymbol{S}_i^{(t)}\right)^{\top}\boldsymbol{S}_i^{(t)}\right) + \lambda + \sum_{i=1}^{B_t}\alpha_i^{(t)}}{\lambda + \sum_{i=1}^{B_t}\alpha_i^{(t)}}$$

$$\leq \frac{\sigma_{\max}\left(\boldsymbol{X}_t^{\top}\boldsymbol{X}_t\right) + \lambda + \sum_{i=1}^{B_t}\alpha_i^{(t)}}{\lambda + \sum_{i=1}^{B_t}\alpha_i^{(t)}}$$

$$\leq \frac{\sigma_{\max}\left(\boldsymbol{X}_t^{\top}\boldsymbol{X}_t\right) + \lambda}{\lambda}$$

$$\leq \text{cond}\left(\boldsymbol{X}_t^{\top}\boldsymbol{X}_t + \lambda\boldsymbol{I}\right),$$

which concludes the proof. □

## H    OPEN ACCESS TO DATA

MNIST: `http://yann.lecun.com/exdb/mnist/`

## I    EXPERIMENT OF ERROR PARAMETER

We evaluate the performance of the proposed method by varying the error parameter $\epsilon$ on the real-world dataset MNIST. The experimental setup is consistent with Section 5.3. Specifically, we set $\epsilon = 200, 1000,$ and $8000,$ and record the spectral norm error $\|\boldsymbol{A}_t^{\top}\boldsymbol{A}_t - \boldsymbol{S}_t^{\top}\boldsymbol{S}_t\|_2$, regret, and running time.

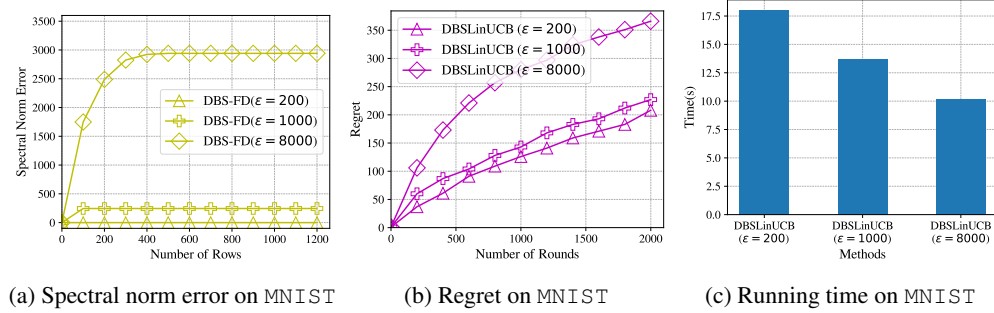

(a) Spectral norm error on MNIST   (b) Regret on MNIST   (c) Running time on MNIST

Figure 5: (a), (b), (c): The spectral norm error, cumulative regret and total running time w.r.t the error parameter $\epsilon$ on MNIST

From Figure 5, we observe that increasing the error parameter $\epsilon$ results in higher spectral error and regret but reduces computational time, which aligns with the theoretical results. Furthermore, in practical applications, it is unnecessary to set $\epsilon$ too small. As shown in Figures 5a and 5b, the performance with $\epsilon = 200$ is comparable to that with $\epsilon = 1000$. This is because $\epsilon$ serves as an upper bound on the spectral error, and appropriate sketching operations have minimal impact on overall performance.

