# OpenReview forum: "Matrix Sketching in Bandits: Current Pitfalls and New Framework"
_ICLR.cc/2025/Conference — Submitted to ICLR 2025_

### Official Review · Reviewer_MFSi · 2024-11-04

**Soundness:** 3
**Presentation:** 1
**Contribution:** 2
**Rating:** 5
**Confidence:** 4

**Summary:**

The authors proposed a new sketching method called dyadic block sketching. This sketching method does not need a fixed sketch size as other sketching methods do, and can adaptively adjust the sketch size based on the error. Using this Dyadic Block Sketching, the authors proposed a bandit algorithm called DBSLinUCB which modifies the traditional OFUL algorithm a bit and achieves an improved regret bound compared to the traditional method of [Kujborskij et al., 2019].

**Strengths:**

The authors proposed a novel and flexible algorithm, DBS, which optimizes its sketching size as the number of samples increases. Adding a doubling trick to the sketching will not be easy, so **if their result is true**, I want to give some points on the novelty, soundness, and significance of this algorithm.

Not only that, they also actually improved the regret bound of the bandit algorithm from the previous work [Kuzborskij et al.., 2019] to a more flexible result. Previously, the result relied on the term called 'spectral error', which is relatively vague and depends on a fixed sketch size. According to their literature search, there was almost no improvement after [Kuzborskij et al., 2019] for 5 years, and they finally made a more flexible and tighter result than before, **if their results are true.**

**Weaknesses:**

Overall, because of clarity issues, I cannot trust the soundness of their result now.

1) Their presentation about the lower bound is poor. I don't understand what the authors want to say in Theorem 1. Please check the 'Questions' section below.

2) Also, Algorithm 1 is the core of this paper, but it is very hard to understand from many perspectives and is not clearly written. Please check the 'Questions' section below.


3) Clarity problems

3-1) Figure 1 - It is really strange that the regret is much larger than the number of timesteps. After I checked the experimental details, they set the arm set $a \sim N(0,I_d)$ and this is the reason why they had a large regret. Also, this means they reported only one experiment, not the average result of repetitions (If not, all their regret should fluctuate a lot). I know that experiments are not the main focus of bandit papers, but this is not a good tradition.

3-2) In Lemma 1, it would be great if they could add the 'size' of each $X_i \in \mathbb{R}^{n_i \times d}$ so that the readers could understand how the submatrices look like. I was a bit confused in Lemma 1 since they didn't define the concatenation $[X_1, X_2]$ before.
2-3) In Section 2.1, they should add their SVD computations, or mention their efficient way of updating parameters $S^{(t)}$ and $M^{(t)}$

4) It seems like it is an improvement from the previous work [Kuzborskij et al., 2019], but I doubt that their result has $(R+L\sqrt{\lambda})$ term in their result. The LinUCB algorithm has a regret bound of $Rd\sqrt{T}$. The form looks quite problematic if $R$ and $L$ are in different scales. I know that the previous work also had this issue, but it is still true that this kind of (variance + arm set scale) is not a good thing in practice, especially when $R<<H$. See Questions section for my doubt on $\lambda$.

**Questions:**

1) Several things I can't understand in Theorem 1

1-1) If I understood correctly, the arm is drawn randomly from a 'fixed' arbitrary distribution over a set of $r$ arms. Now in this case, what is the point of the sketching? The regret above (by Kuzborskij et al) is the regret of the sketch-based linear bandit. This fixed random sampling is not the algorithm by Kuzborskij et al (2019). What is the connection?

1-2) Also, no algorithm could achieve a 'square regret' $\Omega(T^2)$. It is impossible since the worst possible regret is $T$. I can barely guess that the authors want to mention the 'upper bound' of the expected regret using FD is $\Omega(T^2)$, but it is not super surprising, since it is an 'upper bound'. For example, in $K$-armed bandit, the worst case regret bound is of $\sqrt{KT}$, and if $K=T^2$, the regret 'upper bound' is $T^{3/2}$ but it is not surprising since it is just an upper bound.

2) I failed to understand Algorithm 1. Here are the issues I found.

**2-Major**

- If I understood correctly when the algorithm updates, the B[i].length should increase. The term 'rank' is the block's rank, actually. How could a rank be larger than the length? It feels like for $d_1 \times d_2$ dimensional rank $r$ matrix, you are saying 'when $d_1 < r$'.
- What is the second condition in line 276, 'The sum of sketch rows stored in blocks should be less than $d$'? Rows are vectors and $d$ is a scalar.
- Similar to this question, How could a block contain the size $\epsilon l_0$ while doubling the length? In my intuition, each $x_t$ corresponds to the actions chosen by the LinUCB algorithm. Whatever your sketch is, you add $B[i].size +=\|\|x_t\|\|^2$. How can you make the length be doubled while maintaining the same B[i].size for each block?
- The authors mentioned three key invariants that should be maintained - (B[i].length>rank, $B[i].size< l_0 \epsilon$, and the strange condition above). However, in line 8 of Algorithm 1, they used 'AND' logic. Is it right?

**2-Minor**

- $l_0$ is 'sketch size' and you are using the term $B[i].size$ differently. It is very confusing.
- Also the term 'rank' is very confusing. I know they defined rank around line 317, but it's hard to find what it is.
- What is Line 12? The authors should 'define' how each sketching algorithm works, or at least direct readers to a certain part.
- At least use a different font or something for Sk.
- Also, I was about the point out the computational inefficiency of lines 16-18 of Algo 1. I just found that you mentioned in lines 324-326, but it is not intuitive at all. The main point that this paper sells is the computational efficiency, and this part looks like updating all the sketch matrices for each timestep which is not efficient.

3) Also, in Section 2.1, I think they should add some remarks about the rank-$l$ SVD. This algorithm needs rank-$l$ SVD for each time (For Eq. (3), right?) and they intentionally omitted the cost in lines 157-161. As the previous work did, they should add a remark to avoid any doubts about their honesty.

4) The result in Theorem 3 seems much more intertwined in terms of $\lambda$, much more complicated than the previous work of Kuzborskij. Can authors state whether their results are better than Kuzborskij or not in terms of $\lambda$?

5) How large is your error parameter? Is there no assumption needed on $\epsilon$? As we all know, most of the researchers use the character $\epsilon$ as a 'very small quantity smaller than 1'. However in this work, in their experiments, they used $\epsilon = 2000$, much larger than their dimension $d=100$. Are you sure that this $\epsilon$ does not hide any important terms, such as dimensionality?



Overall, I encourage the authors to further refine their paper to bring out its full potential if they believe it has significant contributions to offer.

---

> ### Author Response · Authors · 2024-11-18
> **Response to Reviewer MFSi (1)**
>
> **Response to Reviewer MFSi**:
>
> Thanks for your detailed review of our work!  We sincerely appreciate the time and effort you have invested in reviewing our paper. Your suggestions regarding the presentation and experimental setup have been incredibly valuable to us. However, we have observed that some of your claims, specifically in **W4** and **Q2**, appear to be based on misunderstandings or incorrect assumptions. We hope the clarifications provided below will help facilitate a more balanced assessment of our work.
>
> ---
>
> **W1, Q1:** *About the lower bound*
>
> * Kuzborskij et al. (2019) did not impose assumptions on the environment in their regret analysis. In contrast, we assume that the environment determines the chosen arm $x_t$, which follows a distribution over $r$ orthonormal vectors. This assumption is reasonable, as the environment can influence $x_t$ regardless of the policy used by the algorithm. The sketching step introduced by Kuzborskij et al. (2019) remains applicable, as it represents an efficient policy for selecting the arm, independent of the environmental factors.
>
> * We appreciate the reviewer pointing out the typo $\Omega(T^2)$. As the reviewer noted, we would like to clarify that the 'upper bound' of the expected regret using FD may be linear. Previous work provides a regret upper bound of $(1+\Delta_T)^{\frac{3}{2}} \sqrt{T}$, treating the spectral error term $\Delta_T$ as a constant. However, we have found that $\Delta_T$ is strongly influenced by the environment and may scale linearly with $T$. This means that using the sketching method can be risky, as the 'upper bound' of regret may fail to guarantee sublinear growth. This risk is even more pronounced than in the $K$-armed bandit example, because while $K$ is known before learning, we have no prior knowledge of $\Delta_T$ prior to the learning process.
>
> ___

---

> > ### Author Response · Authors · 2024-11-18
> > **Response to Reviewer MFSi (2)**
> >
> > ---
> >
> > **W2, Q2:** *Algorithm 1*
> >
> > **Q2-Major:**
> >
> > The reviewer seems to have misunderstood our definition of the length of each block. We kindly invite the reviewer to refer to lines 268 and 269. **We define the length of each block as the sketch size stored within the block, which remains a fixed value when updating the block.**
> >
> > * The value of $\text{B}[i].\text{length}$ should only increase when the size of the active block exceeds a certain threshold. For a $d_1 \times d_2$ matrix with rank $k$, where $d_1$ increases over time, we use a sketch with sketch size $\text{B}[i].\text{length}$ to approximate this matrix. **The relationship between the block length and $r$ is ambiguous because $r$ is unknown prior to setting the length.**
> >
> > * We acknowledge that there is a discrepancy in this description. For clarity, we revise it to: "The total number of sketch rows stored in blocks should be less than $d$." The motivation behind designing this invariant is straightforward. Since the non-sketched version stores a matrix of size $d^2$, we aim to align the worst-case scenario of our method with that of the non-sketched version.
> >
> > * As clarified in the previous point, the length of each block corresponds to the sketch size of the associated sketch, which is a **constant**. Note that the size of the block is updated by $\text{size} += \|x_t\|_2^2$, which is independent of the block length (since the length is set when building a new block). Therefore, it is certainly possible to double the length while maintaining the same $\text{B}[i].\text{size}$ for each block.
> >
> > * We double-checked the pseudocode and confirmed that using "AND" is correct. However, we also found that when the length of the active block exceeds the rank of the block, the constraint on the block size can be relaxed. For clarity, we revise the third invariant to: "The size of an inactive block $\text{B}[i]$ should be less than $l_0 \varepsilon$." The reviewer also mentioned that our third invariant appears somewhat unusual. To provide some intuition, we would like to explain that in order to apply the doubling trick, we must ensure that the error is reduced exponentially. Note that the upper bound on the spectral error is $\frac{\|A\|_F^2}{l}$, so it is essential to maintain a bounded Frobenius norm while doubling the length.
> >
> > **Q2-Minor:**
> >
> > * Thank you for pointing out the typos and the inappropriate notations. We will address these issues in the revised version.
> >
> > * We use the term "rank" to indicate whether the streaming sketch processes a matrix with rank smaller than the sketch size. This detection is straightforward in the streaming algorithm. For example, in FD, we compute the shrinking value $\sigma_l$. When $\sigma_l = 0$, it directly indicates that the matrix's rank is less than the sketch size $l$.
> >
> > * We will provide additional intuition for the reviewers regarding lines 16–18. To help readers understand that each inactive block corresponds to a different sketch size and error, we cyclically merged the inactive blocks in these lines. However, this step is actually unnecessary. Since inactive blocks are not updated, we can fully store the merged results of all inactive blocks. In the practical implementation, our algorithm stores only two pairs of matrices: one for the inactive blocks and one for the active blocks. Therefore, only **one merging** of the inactive and active blocks is required during the merging process.
> >
> > ---
> >
> > **W3.1:** *Figure 1*
> >
> > Thank you for pointing out the shortcomings in the experimental setup. Following relevant works, we randomly permuted the synthesized dataset $K$ times and averaged the results. We have updated the experimental results, which are now presented in Figure 1.
> >
> > ---
> > **W3.2, W3.3:** *Definition and introduction*
> >
> > Thank you for pointing out the missing definitions and introductions that caused confusion. We will address these unclear sections in the revised version.
> >
> > ---
> >
> > **W4, Q4:** *The variance and arm set scale*
> >
> > It appears that the reviewer believes the regret of the LinUCB algorithm does not exhibit the structure of variance and arm set scaling. However, as noted in previous work, the non-sketched version, OFUL [1], has a regret bound of $O\left((L\sqrt{\lambda} + R\sqrt{d}) \sqrt{Td}\right)$, which explicitly reflects both variance and arm set scaling. The LinUCB method treats $L$, $R$, and $d$ as constants, focusing on the dominant term $T$, meaning variations in $R$ and $L$ do not impact the regret’s order.
> >
> > Our result matches Kuzborskij’s in terms of the $\lambda$ because the dyadic block sketching technique does not alter the regularization parameter. We did not claim any improvement to the term $\lambda$, as it is generally considered a small constant in related work.

---

> > > ### Author Response · Authors · 2024-11-18
> > > **Response to Reviewer MFSi (3)**
> > >
> > > ---
> > >
> > > **W3:** *SVD step*
> > >
> > > We did not intentionally omitt the cost in lines 157-161. We refer the reviewer to Remark 1 (Efficient Implementation), where we discuss the inefficiency of per-step SVD and propose a more efficient implementation by doubling the space, as detailed in Appendix C. For clarity, we will direct readers to this section in the revised version.
> > >
> > > ---
> > >
> > > **W5:** *Error parameter*
> > >
> > > The error parameter $\epsilon$ serves as an upper bound for the error $||S^{\top}S - A^{\top}A||_2$ introduced by the matrix sketching process. From this perspective, $\epsilon$ is more closely related to the Frobenius norm of the matrix than to its dimensionality. In synthetic data experiments, we observed that sketching can lead to significant spectral loss, as the spectral norm error increases rapidly (see Figure 3). In such cases, even with a large $\epsilon$, the method reverts to a non-sketched approach because the heavy spectral tail results in linear regret. In real-world datasets, where the matrix properties are more favorable for sketching, we can set $\epsilon$ to a value smaller than $T$. We have uploaded a revised version of the paper and invite the reviewer to review our experiment on the error parameter $\epsilon$ in Appendix I.
> > >
> > > ---
> > >
> > > Thank you again for your valuable feedback. If you have any other questions, we are looking forward to more conversations.
> > >
> > > ---
> > >
> > > **Reference**
> > >
> > > [1] Yasin Abbasi-Yadkori, Dávid Pál, and Csaba Szepesvári. Improved algorithms for linear stochastic bandits. Advances in neural information processing systems, 24, 2011.

---

> ### Comment · Reviewer_MFSi · 2024-12-02
>
> Thank you for your sincere rebuttal, but I still feel suspicious.
>
> Q1, W1) If I understood your sentence correctly, in your setting, the environment is acting 'adversarially' that makes your algorithm choose a certain arm' $x_t$ which is a distribution over $r$ orthonormal distribution. Could the authors present one example environment such that the algorithm is forced to sample $x_t$ over $r$ vectors and induce extreme regret? According to Appendix A, I cannot find your specific adversarial environment which forces the learner to choose $x_t$ as $r$ orthonormal distribution.
>
> I also believe addressing these points will take a lot of work, which is not a level of a minor revision, but a clear rewriting, already with this point.

---

> > ### Comment · Reviewer_MFSi · 2024-12-02
> >
> > W2 , Q2) I want to say I keep trying to understand your algorithm, and now I start to understand your direction a little bit. Now if I understood correctly, the $B[i].length$ is the 'limitation' for each block, and 'rank' is the number of meaningful vectors in that block, right? Then $B[i].length$ should be eventually smaller than the number of sketched rows stored in the block. In that sense, I recommend authors change the notation from length to something else.
> >
> > Also for your AND notation, I agree that the authors are right, it should be AND. If not, the learner will always stop stacking sketch blocks exactly after $\epsilon l$.
> >
> > After understanding the blocking things, now I see the points.
> >
> > W4) The point is that, in linear bandit problems, $d$ and $T$ are the most important factors for the regret analysis. Yes, there's S(the upper bound of the reward) term in the regret of LinUCB, but it is not with dimensionality (only with $\lambda$), so it is a minor term. Overall, LinUCB result is mainly $Rd\sqrt{T}$. On the other hand, in your work, your result is something like $(LR+S)d\sqrt{T}$ (I know there are tons of other serious terms in addition) which means worse. I can't say this is a major drawback, but I want to say the rebuttal of the authors about this point was not the thing that I wanted.
> >
> > Overall, I believe there are lots of things that this paper needs to be revised, but I barely understand the thing that the authors want to say I will increase my score to marginally below the threshold.

---

> > > ### Author Response · Authors · 2024-12-03
> > > **Response to Reviewer u3gg**
> > >
> > > We greatly appreciate the reviewer's interest in understanding the work! Thank you for your continued investment.
> > >
> > > ---
> > >
> > > **Q1, W1:** *Theorem 1*
> > >
> > > We provide a simple example environment to induce the extreme regret as discussed in Theorem 1. In the round $t$, the environment first sample a vector $a_i$ from the orthonormal vectors $A$ (as discussed in Appendix A). Then, the environment provides the available arm set $\mathcal{X}_t = \{c_1a_i,...,c_pa_i\}$ to the learner, where $c_1,...,c_p$ are the random scalar multiple. Note that the scalar multiple **will not affect the rank of matrix**, therefore the learner will forced to choose the arm over orthonormal vectors.
> > >
> > > ---
> > >
> > > **Q2, W2:** *Algorithm 1*
> > >
> > > We are pleased that our clarification has helped the reviewers better understand our definition and the context of the block. To enhance the clarity of the paper, we have updated the results discussed with the reviewers in the newly uploaded PDF.
> > >
> > > Additionally, we would like to provide further details regarding the variable 'rank'. For a matrix $ A $ with rank $ k $, when approximating it with a sketch size $ l < k $, we can only know that $ l < k $, but not the exact value of $ k $. When $ l \approx k $, matrix sketching can accelerate the algorithm with almost no loss of spectral information (though this is not realistic for single-scale sketching, as the value of $ k $ is unknown in advance). 'rank' is essentially a **Boolean variable**, and as we cautiously increase the matrix sketch size to $ l \approx k $, this variable signals to the algorithm that the optimal sketch approximation has been reached.
> > >
> > > ---
> > >
> > > **W4** *The variance and arm set scale*
> > >
> > > We thank the reviewer for pointing out that our previous rebuttal did not fully clarify the confusion. After a thorough review of the theoretical proofs in our work, we discovered that the issue stems from our **simplification of certain terms** (a similar step was also used in previous works).
> > >
> > > We invite the reviewers to review our proof in Appendix E. Specifically, on **Line 1086**, we derived the following expression:
> > > $$
> > > Regret_T \le \frac{L}{\sqrt{\lambda}}\cdot\sqrt{T}\cdot\left(1+\frac{\epsilon}{\lambda}\right)\cdot\left(d\ln\left(\frac{1+\epsilon}{\lambda}\right)+2l_{B_t}\cdot\ln\left(1+\frac{TL^2}{2l_{B_{T}}\lambda}\right)\right)
> > > \cdot
> > > \Bigg(R\sqrt{1+\frac{\epsilon}{\lambda}}\cdot\sqrt{2\ln\frac{1}{\delta}+d\ln\left(1+\frac{\epsilon}{\lambda}\right)+2l_{B_T}\cdot\ln\left(1+\frac{TL^2}{2l_{B_{T}}\lambda}\right)}
> > > +H\sqrt{\lambda}\left({1+\frac{\epsilon}{\lambda}}\right)\Bigg)
> > > $$
> > >
> > > This result follows **the same structure** as the LinUCB result, where the $ H $ term (the upper bound of the reward) involves only $ \lambda $, and the $ R $ term with the dimensionality. To simplify this result, we (as well as previous works) merge the two terms and retain the higher-order term.
> > >
> > > ---
> > >
> > > Thank you again for your feedback. If you have any other questions, we are looking forward to more conversations.

---

### Official Review · Reviewer_u3gg · 2024-11-06

**Soundness:** 2
**Presentation:** 2
**Contribution:** 2
**Rating:** 6
**Confidence:** 2

**Summary:**

This work explores the limitations of current sketching techniques in online learning, particularly within linear bandit settings. Existing methods using matrix sketching suffer from high spectral error, leading to potential linear regret when the covariance matrix's spectral tail does not decay rapidly. The authors propose a novel approach, Dyadic Block Sketching, which adaptively adjusts sketch size to control spectral error. This new technique aims to achieve sublinear regret without prior knowledge of the covariance matrix, presenting a potentially generalizable framework for improving efficiency and regret bounds in sketch-based linear bandits.

**Strengths:**

- The topic of balancing computational efficiency with accurate regret minimization in high-dimensional settings is of broad interest
- The writing is mostly clear

**Weaknesses:**

- Magnitude of the technical novelty and practical impact of the contributions

**Questions:**

- Did you run any additional experiments on more diverse real-world datasets to better investigate the method's adaptability to varying spectral properties? If so, what did you observe?
- Given the increased complexity of managing blocks and adaptive sketch sizes, how does Dyadic Block Sketching compare in terms of computational complexity to simpler, single-scale sketching approaches in real-world applications?
- The paper suggests improved regret bounds when favorable conditions are met. Could you elaborate on how practical these improvements are in some examples of real-life scenarios?

---

> ### Author Response · Authors · 2024-11-18
> **Response to Reviewer u3gg**
>
> **Response to Reviewer u3gg**:
>
> Thanks for your detailed review of our work! We appreciate your insights and have addressed your concerns below.
>
> ---
>
> **Q1:** *Additional experiments*
>
> The MNIST dataset is a widely used real-world benchmark in related works. Its high dimensionality and uneven spectral distribution make it particularly well-suited for studying sketching problems. Thus, we believe that our current real-world experiments provide sufficient evidence to support our conclusions. However, we are open to incorporating additional, more diverse real-world datasets in future work as needed.
>
> ---
>
> **Q2:** *Complexity*
>
> In real-world applications, relying on single-scale sketching is risky, as the properties of the matrix are often unknown. Although the complexity of our method may increase during the learning process, it has the ability to adapt to potential low-rank structures (as outlined in Section 3.3). This means that when the spectral tail of the matrix is small, our method automatically adjusts to the optimal sketch size, whereas single-scale sketching requires prior knowledge of the matrix properties before learning .
>
> ---
>
> **Q3:** *Improved regret bound*
>
>
> In most online learning literature, hyperparameters must be carefully tuned to ensure a low regret upper bound. However, single-scale sketching introduces an uncontrollable term $\Delta_T$ into the regret bound. Our method aligns with traditional online learning approaches by relying only on a preset constant $\epsilon$ in the regret. As discussed previously, under favorable conditions (e.g., a low spectral tail, which is common in many high-dimensional datasets), our method ensures a sublinear regret bound while maintaining computational efficiency.
>
> ---
>
> Thank you again for your valuable feedback. If you have any other questions, we are looking forward to more conversations.

---

> > ### Comment · Reviewer_u3gg · 2024-11-26
> >
> > Thank you for answering my questions. I maintain my score.

---

### Official Review · Reviewer_3oww · 2024-11-07

**Soundness:** 3
**Presentation:** 2
**Contribution:** 3
**Rating:** 6
**Confidence:** 3

**Summary:**

This paper studies matrix sketching within the context of linear bandits, aiming to reduce time complexity.

The authors demonstrate that Frequent Directions (FD), a previously utilized matrix sketching method, could lead to linear regret. They introduce a dyadic block sketching method where the sketch size is adaptively adjusted based on the parameter $\epsilon$. The main theorem asserts that the proposed algorithm achieves favorable regret bounds, and experimental results highlight its effectiveness and efficiency.

**Strengths:**

The authors demonstrate that Frequent Directions (FD), a previously utilized matrix sketching method, could lead to linear regret. They introduce a dyadic block sketching method where the sketch size is adaptively adjusted based on the parameter $\epsilon$. The main theorem asserts that the proposed algorithm achieves favorable regret bounds, and experimental results highlight its effectiveness and efficiency.

**Weaknesses:**

However, there are several weaknesses:

1. The comparison with previous work is overly simplistic and lacks sufficient detail. In previous work, $l$ represents a trade-off between time complexity and regret. This work uses $\epsilon$. The authors should provide a comprehensive analysis of the trade-off (including time complexity at each step, memory costs, assumptions, and regret) to elucidate the advantages of the new method.
2. Although a regret lower bound is established for the previous algorithm, it is unclear how the proposed algorithm performs under the worst-case scenario constructed.
3. The algorithm searches for parameters in the experiments. However, the main theorems do not hold for these parameters. It would be beneficial if the authors could also present experimental results that consider scaling the parameters.

Additionally, in Theorem 1, the stated regret is $T^2$, which is unusual since the worst-case scenario typically scales linearly with $T$. Could the authors clarify this discrepancy?

**Questions:**

See weaknesses.

---

> ### Author Response · Authors · 2024-11-18
> **Response to Reviewer 3oww**
>
> **Response to Reviewer 3oww**:
>
> Thanks for your detailed review of our work! We have addressed your concerns and provided clarifications below.
>
> ---
>
> **W1:** *Comparison with previous work*
>
> The review comments focus on the differences between our method and previous work. We invite the reviewers to refer to the following sections of our paper, where we separately discuss the limitations of previous work, as well as the complexity and regret:
>
> * In Section 2.2, we discuss the limitations of previous work. They use a fixed sketch size $l$ to control the complexity of the algorithm, but this can lead to linear regret due to the uncontrollable term $\Delta_T$. While $l$ represents a trade-off between time complexity and regret, it remains risky because the spectral error $\Delta_T$ is not constant and increases rapidly over time.
>
> * In Section 3.3, we discuss the space and amortized time complexity of our method. The per-step complexity is addressed in line 405. The previous method performs ideally when the matrix has favorable properties, such as low rank. In Corollary 1, we show that under these conditions, the complexity of our method aligns with the optimal single-scale sketch complexity. Unlike the previous method, which requires prior knowledge of the matrix (e.g., rank $k$) to achieve optimal complexity, our method does not rely on any prior knowledge.
>
> * In Theorems 3 and 4, we discuss the corresponding regret bounds and compare our method with previous work. The main advantage of our approach is that it fixes the order of the regret bound while adjusting the complexity. In contrast to previous methods that fix the complexity but lead to an uncontrollable regret bound, our method provides a more practical and reasonable solution for real-world applications.
>
>
> ---
>
> **W2:** *Worst-case scenario*
>
> In Theorem 1, we demonstrate that when the sketch size $l$ is smaller than $r$, the result is linear regret due to excessive loss of spectral information during sketching. Previous algorithms cannot adapt to this worst-case scenario because the sketch size $l$ is fixed prior to learning, while $r$ is unknown. In contrast, our method dynamically adjusts the sketch size during learning. As shown in Corollary 1, our method maintains an active block with a sketch size $l_{B_t} > r$ after some rounds, thereby achieving sublinear regret even in the worst-case scenario.
>
> ---
> **W3:** *Scaling the parameters*
>
> The core parameter of our method, $\epsilon$, represents the upper bound of the matrix approximation error. We believe that conducting scaling experiments on this parameter will enhance the insights provided in our article. We have uploaded a revised version of the paper and invite the reviewer to review our experiment on the error parameter in Appendix I.
>
> ---
>
> **W4:** *Theorem 1*
>
> We sincerely thank the reviewer for pointing out this typo. Previous work treats the spectral error $\Delta_T$ as a constant term in the regret upper bound $O\left((1 + \Delta_T)^{\frac{3}{2}} \sqrt{T}\right)$. However, we found that $\Delta_T$ is likely not constant but increases with $T$, making the previous method risky. The motivation behind Theorem 1 is to show that when the environment produces a matrix with a heavy spectral tail, the spectral error $\Delta_T$ becomes dependent on $T$, leading to a linear regret upper bound. We will revise the expression of Theorem 1 in the updated version for clarity.
>
> ---
>
> Thank you again for your valuable feedback. If you have any other questions, we are looking forward to more conversations.

---

> > ### Author Response · Authors · 2024-12-03
> > **Summary and Looking Forward to Further Discussions**
> >
> > Thank you again for your great efforts and valuable comments. As the author-reviewer discussion phase comes to a close, we look forward to any additional feedback you may have. Below, we summarize our previous responses for your convenience:
> >
> > **W1**: We have thoroughly discussed the time and space complexity, error, and regret bounds of the algorithm in multiple sections of the paper. Additionally, we have provided a comparison with previous work.
> >
> > **W2**: We demonstrated that our algorithm can avoid the worst-case scenario presented in Theorem 1, as our sketch size can be dynamically adjusted (in contrast to single-scale sketching, where the sketch size is fixed).
> >
> > **W3**: We have included additional experiments that scale the parameters.
> >
> > **W4**: For clarity, we have revised the expression of Theorem 1 in the updated version.

---

> > > ### Comment · Reviewer_3oww · 2024-12-03
> > >
> > > Thank you for your response; it addresses most of my concerns, and I have updated my score accordingly.
> > >
> > > I do have an additional question: your sketch size can be dynamically adjusted based on whether it is the worst-case scenario. Is there any metric or discussion to evaluate how effectively this adjustment is performed?
> > >
> > > For example, could you include a discussion about the potential implications if the scaled size is $\ell_{B_{t}}^2$ (larger than yours)? Specifically, for certain instances, might this lead to a linear regret? I would appreciate seeing a more detailed and reasonable discussion on this aspect.

---

> > > > ### Author Response · Authors · 2024-12-03
> > > >
> > > > Thank you again for your great efforts and valuable comments.
> > > >
> > > > We are confident that the adjustments we have made are effective, as the matrix sketching method is capable of **capturing the low-rank structure**. We kindly invite the reviewer to refer to the following discussion in the paper:
> > > >
> > > > 1. In **Theorem 1**, we demonstrate that sketching results in linear regret when the sketch size $ l < r $. However, when $ l > r $, even a single-scale sketch can prevent linear regret, as the spectral error $ \Delta_T = 0 $.
> > > >
> > > > 2. In **Corollary 1, we show that the last active block will have a sketch size close to $ k $ (which equals $ r $ in Theorem 1)**, because the amortized cost is dominated by the active block. **The crucial technical point is the variable $ rank $ (defined on Line 316).** If the active block has a sketch size larger than $ k $, the $ rank $ variable will prevent the learner from constructing new blocks, ensuring that the spectral error $ \Delta_T = 0 $ in the current active block.
> > > >
> > > > Additionally, we are uncertain about the reviewer's mention of the term 'scaled size.' Is this intended to represent the potential rank of the streaming matrix?

---

### Official Review · Reviewer_RQYf · 2024-11-10

**Soundness:** 3
**Presentation:** 2
**Contribution:** 2
**Rating:** 5
**Confidence:** 3

**Summary:**

The authors consider stochastic linear bandits when the dimension $d$ is high enough that per-round computational complexity is of concern to the user.  Traditional methods have high per-round computational complexity with respect to the dimension size $d$ due to covariance matrix estimators used.  Sketching based variants have been proposed to alleviate the computational complexity, but require pre-selecting the sketch size $l$ which can lead to high-regret (there is a regret bound term related to error from sketching (with size $l$) that can exhibit horizon dependence).  The authors propose a new sketching method that adaptively selects the sketch size subject to a parameter controlling the sketching error.  They derive regret bounds and show experiments demonstrating that it can empirically achieve a superior tradeoff of regret and computation.

**Strengths:**

- The general topic of improving trade-offs in per-round computation and regret in online learning is important.

- The authors clearly lay out an important issue with standard stochastic linear bandit methods with the dimension is large enough that per-round computation can be a concern.

- The proposed method is interesting, using a decomposition property to control the overall spectral norm error and then carefully designing block sizes adaptively.  It is integrated using standard matrix sketching methods (FD and RFD) to form new LinUCB variants for which the authors obtain regret bounds.

- The presentation overall is good.

- Experiments are included on synthetic and real-world data, showing that the method can perform competitively with non-sketching methods while reducing per-round computation significantly.

**Weaknesses:**

### Major

- My biggest concern is with regards to prior works.  There is a brief related works discussion on sketching (no recent works) and no mention of similarities or differences in the methodology section.  Is this Dyadic Block Sketching technique completely new?  Three classes of matrix sketching methods are described none of which seemed (to me) to be particularly similar.

    - However, there is at least one (slightly more recent) paper that is not cited but sounds fairly relevant: a 2016 sigmod https://dl.acm.org/doi/abs/10.1145/2882903.2915228 paper which proposes a “Dyadic Interval framework” that sounds similar—it is also based on the decomposability property, it also involves specifying an error hyper-parameter, has a logarithmic number of ‘levels’, and there is a check of a L2 norm based block size that triggers updating; they also combine it with the “Frequent Directions” method.

    - The citations included in that section are old, with the most recent paper in that section of related work being a 2014 review paper.



- Another concern I have is with the role of $\epsilon$.  I think it is quite interesting to go from fixing a sketch size and possibly incurring large regret to fixing an error bound in part of the estimation scheme.  Given there is a different hyperparameter,  it would have been better to have more discussion and also some experiments showing how the choice of $\epsilon$ impacts the spectral norm errors, the regret, and the computation.  In the experiments a seemingly large constant $\epsilon=2000$ is used for two sets of experiments, $\epsilon = 1000$ in another, but I do not have a sense where those came from.  For sketch size, by contrast, the user knows $d$ and at least for a target computational budget could probably fairly easily select a sketch size that would fit well within that (with the caveat being of course it might incur large error).

    - 5.1 Matrix approximation experiment – only one value of $\epsilon$ is used (also computation time and sketch sizes method used are not reported for that $\epsilon$);

    - 5.3 experiment on MNIST, “In contrast, DBSLinUCB matches the performance of OFUL by adaptively adjusting the sketch size to the near-optimal value of l = 200 while being significantly faster than OFUL.” empirical regret is shown so clearly is close to OFUL and time is shown for this experiment so is clear is close to SOFUL(l=200), which is suggestive sketch sizes would be similar (around 200) but did  you actually check that?  Or are you basing that on the total run-time?

    - in experiment 5.3, the $\epsilon$ was changed compared to previous experiments? Why?  How was this value chosen?





### Minor

- the computational complexity (and relatedly realized sketch sizes) are not reported.  In the first experiment the spectral norm error is shown (so tradeoff FD makes is clear) but no mention of the extent to which computation time is traded off.

- in section 5.1, is there a reason FD does better for small number of rows than DBS-FD?

- theorem 1’s statement is poorly written for a formal theorem.  It also contains a notable typo “$\Omega(T^2)$” for regret bound

**Questions:**

(several questions are included as part of concerns mentioned above)

---

> ### Author Response · Authors · 2024-11-18
> **Response to Reviewer RQYf**
>
> Thanks for your detailed review of our work! We have addressed your concerns and provided clarifications below.
>
> ---
>
> **W1:** *Prior work*
>
> * The decomposability of matrix sketches is a well-known property in streaming algorithms and has been widely used in the literature (including the “Dyadic Interval framework” mentioned by the reviewer). While our method shares this property, there is a fundamental difference compared to traditional streaming methods. We invite the reviewer to refer to the following sections of our paper for clarification.
>
>     In Remark 2 (line 358), we demonstrate that our method reverts to a non-sketched version in the worst-case scenario, which means it does not rely on space-bounded assumptions. In contrast, traditional streaming methods require a fixed space usage (e.g., the “Dyadic Interval framework” stores at most $l$ rows to maintain a sliding window sketch). Additionally, the Sigmod paper considers a matrix sketching algorithm over the sliding window model, which is unrelated to our work, and we did not cite it.
>
> * The citations regarding matrix sketching techniques are older because these methods have been well-studied for over a decade. For example, the Frequent Directions method was proven to be optimal by Liberty (2013).
>
> ---
>
> **W2:** *Error parameter $\epsilon$*
>
> We appreciate the reviewer's insights on the error parameter $\epsilon$, and we glad to provide some discussions and experiments to help readers better choose $\epsilon$. We have uploaded a revised version of the paper and invite the reviewer to review our experiment on the error parameter $\epsilon$ in Appendix I.
>
> From the theoratical perspective, we should assume $\epsilon$ as a much smaller constant than $T$ to achieve the sublinear regret bound. However, during the actual experimental process, we found that it is most effective to set $\epsilon$ slightly larger than the theoretical value for two main reasons:
> * In some cases, the upper bounds for both components are not tight. For matrix approximation, as shown in Figure 3, the error upper bound of our method can be much larger than the actual error. Similarly, for the arm selection strategy, a certain level of matrix approximation may not significantly impact the choice of a specific arm, meaning that the regret upper bound can be much larger than the actual performance.
> * When $\epsilon$ is set too small, the algorithm's robustness to row norm size deteriorates. For example, when rows with larger norms (such as outliers) arrive, a smaller $\epsilon$ causes the threshold for creating new blocks to decrease, resulting in the algorithm unnecessarily expanding the new blocks. Moreover, we found that reducing $\epsilon$ does not significantly improve algorithm performance, but rather increases time complexity. This is because, at the optimal sketch size, the algorithm’s performance is nearly identical to the non-sketch version. A very small $\epsilon$ only leads the algorithm to quickly revert to the non-sketch version, which limits the potential to optimize time complexity using sketches.
>
> In summary, **we have adopted a simple yet effective criterion for selecting $\epsilon$, which is based on the total number of rounds, $T$.** We recommend choosing a value for $\epsilon$ that is smaller than $T$, which is straightforward since $T$ is typically known in advance. If prior knowledge suggests that the row norms of certain matrices are relatively large, we can even set $\epsilon$ equal to $T$ (as we did in the synthetic experiments).
>
> ---
>
> **W3, W4:** *Matrix approximation experiment*
>
> * In the first experiment, we address the problem of matrix approximation. Note that the upper bound of the error caused by an FD sketch with sketch size $l$ is $\frac{||A||_F^2}{l}$. Since the per-round complexity of our method is dominated by the sketch size of the active block, the computational complexity can be inferred from the slope of the curve. For example, the slope of DBS-FD is initially steeper than that of FD, as the sketch size in the first block of DBS-FD is 16, while the fixed sketch size of FD is 50.
>
> * As mentioned previously, FD performs better than DBS-FD for a small number of rows because, in the initial stages, DBS-FD uses a smaller sketch size (smaller than FD) to approximate the matrix.
>
> ---
>
> **W5:** *Theorem 1*
>
> We appreciate the reviewer pointing out this typo. Previous work treats the spectral error $\Delta_T$ as a constant term in the regret upper bound $O\left((1 + \Delta_T)^{\frac{3}{2}} \sqrt{T}\right)$. The motivation behind Theorem 1 is to demonstrate that when the environment produces a matrix with a heavy spectral tail, the spectral error $\Delta_T$ becomes dependent on $T$, leading to a linear regret upper bound. For clarity, we will revise the expression of Theorem 1 in the updated version.
>
> ---
> Thank you again for your valuable feedback. If you have any other questions, we are looking forward to more conversations.

---

> > ### Author Response · Authors · 2024-12-03
> > **Summary and Looking Forward to Further Discussions**
> >
> > Thank you again for your great efforts and valuable comments. As the author-reviewer discussion phase comes to a close, we look forward to any additional feedback you may have. Below, we summarize our previous responses for your convenience:
> >
> > **W1**: We clarified the fundamental distinction between our work and streaming algorithms, emphasizing that our approach does not impose strict space complexity limitations.
> >
> > **W2**: We provided selection criteria for the error parameter $\epsilon$, which is simple to apply in practice since only the total number of rounds $ T $ needs to be known. Additionally, we have included experiments with various values of $ \epsilon $ to demonstrate the practicality of our selection criteria.
> >
> > **W3, W4**: We addressed the questions regarding the matrix approximation experiments.
> >
> > **W5**: For improved clarity, we revised the expression of Theorem 1 in the updated version.

---

### Author Response · Authors · 2024-11-18

In the uploaded PDF file, we have added the following content:

* We add experiments that consider scaling the error parameter $\epsilon$ in Appendix I.

* We update Figure 1 with average result of repetitions.

---

### Meta-Review · Area_Chair_hVBZ · 2024-12-20

**Metareview:**

Recognizing that existing sketching methods for speeding up linear bandit algorithms perform poorly when the rank of the problem is not small compared to the sketching dimension, this paper presents a method for adjusting the sketching dimension "on the fly" to adapt to the problem rank. While all reviewers saw merit in the paper, the overall novelty and conceptual contribution are somewhat limited compared to other submissions. While nice to have proven, the conceptual message behind the lower bounds provided by the paper is expected, and the approach of adaptively adjusting the sketch requires effort to develop, but does not represent a major new idea. That said, we appreciate the authors' engagement with the reviewers and the relative clarity of the paper, and wish the authors the best of luck submitting to another venue.

**Additional Comments On Reviewer Discussion:**

The authors clarified some technical points (including e.g., a lower bound proof that one reviewer had questions about). Overall though, there was not a major change of opinion following the discussion.

---

### Decision · Program_Chairs · 2025-01-22

Reject